# LEARNING HIERARCHICAL DOMAIN MODELS THROUGH ENVIRONMENT-GROUNDED INTERACTION

## ABSTRACT

Domain models enable autonomous agents to solve long-horizon tasks by producing interpretable plans. However, in open-world environments, a single general domain model cannot capture the variety of tasks, so agents must generate suitable task-specific models on the fly. Large Language Models (LLMs), with their implicit common knowledge, can generate such domains, but suffer from high error rates that limit their applicability. Hence, related work relies on extensive human feedback or prior knowledge, which undermines autonomous, open-world deployment. In this work, we propose LODGE, a framework for autonomous domain learning from LLMs and environment grounding. LODGE builds on hierarchical abstractions and automated simulations to identify and correct inconsistencies between abstraction layers and between the model and environment. Our framework is task-agnostic, as it generates predicates, operators, and their preconditions and effects, while only assuming access to a simulator and a set of generic, executable low-level skills. Experiments on two International Planning Competition (IPC) domains and a robotic assembly domain show that LODGE yields more accurate domain models and higher task success than existing methods, requiring remarkably few environment interactions and no human feedback or demonstrations.

## 1 INTRODUCTION

A central goal in robotics is to enable autonomous agents to operate in open-world environments and solve long-horizon tasks such as setting a table (Sun et al., 2019) or retrieving and stowing packages across a warehouse (Bernardo et al., 2022). Planning is essential in this context, as it produces structured and interpretable plans grounded in explicit domain models, in contrast to opaque VLAs (Firoozi et al., 2025). The domain model describes the world through objects, predicates defining affordances and relations, and operators specifying available actions. However, open-world environments make it impossible to assume a single ground-truth domain model: The relevant objects, relations, and actions differ across tasks and settings. Even if constructing a single, universal domain were feasible, planning within it would be computationally intractable. Hence, learning context-specific domain models on the fly is crucial for explicit, interpretable open-world reasoning. Significant work has addressed learning domain models from demonstrations (Silver et al., 2021a). However, to be practical for open-world settings, domain models must be learned from limited data, since collecting extensive demonstrations or annotations for every new setting is infeasible. Instead, LLMs can facilitate autonomous online-learning of context-specific domain modes by retrieving relevant knowledge. However, their outputs are not reliable: forgotten or hallucinated preconditions and effects may result in domains that are unsolvable, not executable by the robot, or misaligned with the intended goal. Hence, prior works rely on verbose and task-specific skill annotations (Mahdavi et al., 2024) or significant human feedback (Guan et al., 2023) to improve the correctness of LLM-generated domains, but thereby also limit their applicability for online, open-world domain learning.

To address these challenges, we propose LODGE, a fully autonomous and task-generic method for learning domain models. LODGE learns domain models without trusted knowledge sources by enforcing consistency across layers of a hierarchical domain model, between the domain model and a robot simulation, and across interacting LLM and VLM-based modules. LODGE leverages an LLM to generate an initial domain model, which it iteratively refines and adapts in a targeted manner while solving tasks using the domain model in the environment. We show that our method requires remarkably few environment interactions to construct Planning Domain Definition Language

(PDDL) domain models that are more accurate than related work. The difficulty of learning domain models with LLMs increases with the number of operators and predicates (Kambhampati et al., 2024). To address this, we propose to learn hierarchical domain models. LODGE first generates an abstract domain model and plan that it subsequently decomposes into subdomains and plans. Hierarchical decomposition enables partial re-planning of subplans while maintaining alignment across hierarchy levels. Unlike approaches that propose sub-goals (Liu et al., 2025), hierarchical domain models do not suffer from detrimental sub-goal interactions (Kambhampati et al., 2024). To avoid relying on predefined task-specific predicates, we propose an algorithm for inventing new predicates and corresponding classifiers to ground the symbolic state of the environment from the low-level continuous state. We show how the accuracy of learned classifiers can be improved by autonomously optimizing the classifiers' hyperparameters (e.g. tolerances) and by LLM self-refinement based on a dataset of past environment interactions, achieving higher accuracy than related methods. Lastly, we propose a *global recovery module* that detects misalignments between the symbolic state of the environment and the planner's symbolic state. Upon misalignment, it analyzes the past domain learning and trigger a targeted recovery step to restore alignment with the observed symbolic state and recover domain learning. With that, our contributions are as follows:

**Domain learning from planning feedback:** By combining LLMs with predicate classifiers, we detect misalignments between the symbolic planner's state and the observed environment state. With the *global recovery module*, we show how domain operators can be refined in a targeted manner. This proposes the first automated method to learn domain models from few environment interactions without human feedback, enabling use in open-world environments.

**Hierarchical domain models:** We introduce a hierarchical approach to domain learning in which abstract operators are progressively decomposed into reusable subdomains. This enables the distribution of domain model complexity across multiple abstraction levels, which simplifies learning and reduces computational planning complexity.

**Predicate invention and online classifier learning:** We introduce a method to invent new predicates and learn their classifiers from pseudo-labeled environment interactions. This shows that the symbolic state can be extended autonomously, which enables representing and grounding previously unknown aspects of the environment without human feedback.

## 2 RELATED WORK

**Planning with LLMs**  Early work used LLMs to generate plans directly from natural language (Ichter et al., 2023). Liang et al. (2023) proposed hierarchical plan decomposition, enabling LLMs to define and later implement unknown functions. Others explored specifying tasks with formal planning languages (Valmeekam et al., 2023) to reduce natural language ambiguity (Pallagani et al., 2024). While natural language instructions offer flexibility, LLM planners lack correctness guarantees, making plan validation difficult. State-of-the-art models still struggle with simple planning due to limited explicit reasoning (Valmeekam et al., 2023). To address these limitations, hybrid methods combine LLMs with classical planners. One approach uses LLMs to translate natural language instructions into formal goal specifications interpretable by classical planners (Liu et al., 2023). The syntax and semantics of problem definitions can be verified externally (Howey et al., 2004) or via LLM self-critiques (Zhou et al., 2024). Unlike classical planners, LLM planners provide feedback when plans fail (Valmeekam et al., 2023), and plan correctness can be validated externally (Zhou et al., 2024) However, these methods assume a predefined domain model, which is difficult to construct.

**Hierarchical Planning**  Hierarchical planning has been part of AI planning since its early days (Fikes and Nilsson, 1971), motivated by the computational benefits of shorter plans for long-horizon tasks and complex instructions. The core idea is to decompose a high-level action (HLA) into a sequence of lower-level actions, possibly including other HLAs (Marthi et al., 2007). Early languages, such as the *macro-operator* description in (Fikes and Nilsson, 1971) restrict each HLA to exactly one decomposition. Botea et al. (2005) later learn PDDL macros from concrete planning problems by identifying patterns in a plan and wrapping them into a hierarchy level as macros. Allowing multiple decompositions per HLA increases versatility, since multiple sub-plans can typically implement an HLA (Ghallab et al., 2016). Yet, this versatility comes at the cost of complicating the definition of

an HLA's effects, which may vary for different decompositions. Defining accurate HLA effects is therefore crucial (Srivastava et al., 2016), whereby some propose using conditional effects, while others motivate non-deterministic effects to define hard-to-track effects Marthi et al. (2007). Our method focuses on learning hierarchical domain models to distribute the complexity of learning large domain models across multiple steps in the planning process. In particular, we focus on automatically deciding *when to learn a decomposition*. Our formulation of hierarchical domain models relates to hierarchical planning, but with some key differences from hierarchical planning domains. First, we model the hierarchical decomposition as its own planning problem and automatically derive the corresponding problem definition from the high-level operator. This approach offers two key advantages: (i) each hierarchy level uses a smaller, more focused domain model, which is easier to generate than a single monolithic model; and (ii) the state space of a lower-level domain can be a superset of its parent's state, which simplifies high-level operators, especially when tracking complex state changes. Second, within a given hierarchy level, we treat high-level operators in the same way as primitive ones, allowing us to plan for them using an off-the-shelf classical planner. With that, we plan decompositions *online* instead of predefining valid decompositions in the domain model.

**Learning Domain Models with LLMs** Constructing domain models is complex, but crucial for autonomous planning systems. The construction with LLMs is, due to the resulting sample efficiency, particularly promising for open-world domains, but remains underexplored (Tantakoun et al., 2025). Recent work verifies LLM-generated domains by self-critique or classical tools (Smirnov et al., 2024; Ishay and Lee, 2025; Birr et al., 2024; Chen et al., 2024; Ding et al., 2023). Complementary work generates multiple domain candidates and evaluates them using similarity metrics or environment interaction (Yu et al., 2025; Hu et al., 2025). Guan et al. (2023) propose an LLM-based framework to construct PDDL domain models by first translating predefined skills into action definitions, followed by iterative refinement via human feedback. Oswald et al. (2024) focus on reconstructing high-quality PDDL domains from natural language descriptions. Recent work by Mahdavi et al. (2024) avoid human feedback by sampling multiple candidate domains and iteratively refining the one closest to a partially known domain model.

In contrast to these approaches, LODGE learns hierarchical domain models with LLMs and environment grounding, without skill annotations or human feedback. Additionally, LODGE does not rely on partially known domain models and introduces a targeted refinement of models through environment interaction.

**Predicate Invention and Learning Classifiers** Previous work extracts predicates from demonstration data (Keller et al., 2025) or learns predicate classifiers that map the continuous environment state to the planner's symbolic state (Migimatsu and Bohg, 2022). Furthermore, predicates can be infered and grounded by constructing scene graphs (Ekpo et al., 2024; Herzog et al., 2025). Large pretrained models can also ground predicates from images of the scene (Liang et al., 2025; Athalye et al., 2025), or from perception data, such as 6D poses (Shirai et al., 2024). (Liang et al., 2025) propose neuro-symbolic predicates, which combine VLM invocations with symbolic reasoning to ground predicates. Alternatively, LLMs can generate interpretable predicate classifiers in code, e.g., Python, avoiding the need for LLMs during inference (Han et al., 2024). Given a set of learned predicates and grounded transitions, domain operators can automatically be inferred, e.g. with the *cluster and intersect* algorithm proposed by Chitnis et al. (2022)

Our algorithm for learning predicate classifiers combines recent work of Han et al. (2024) and Athalye et al. (2025). We iteratively refine the constructed predicate classifiers based on past environment interactions labeled with pseudo-labels instead of requiring human feedback, and propose two new techniques to refine predicate classifiers based on a pseudo-labeled dataset.

## 3 NOTATION AND PROBLEM STATEMENT

We consider an environment $f$ (real or simulated) with a continuous state $x$, where each object $o \in \mathcal{O}$ is mapped to real-valued vectors, e.g. its 6D pose. An actor can execute a skill $\pi$ in the environment, which is parameterized by a set of variables $\mathcal{V}$ that can be grounded with objects $o_i \in \mathcal{O}$ existing in the environment. We denote a sequence $o_1, \ldots, o_n$ by $o_{1:n}$. Executing $\pi$ on $f$ transitions the environment into the next state $x_{t+1} = f(x_t, \pi(o_{1:n_\pi}))$. The domain model $\mathcal{D} = \langle \Psi, \Omega \rangle$ consists of predicates $\Psi$ and operators $\Omega$. Every predicate $\psi \in \Psi$ encodes a discrete property of the state $x$,

such as *grasps(?obj)* or *stacked(?obj1, ?obj2)*. We distinguish between *state-based* predicates that operate exclusively over the object-centric state, and *state-independent* predicates. Each *state-based* predicate $\psi$ has a corresponding classifier $c_\psi : \mathcal{O} \times \mathcal{X} \rightarrow \{\text{true}, \text{false}\}$. $c_{\underline{\psi}}(x) = c_\psi(o_{1:n_\psi}, x)$ evaluates whether predicate $\psi$ with objects $o_{1:n_\psi}$ (ground atom $\underline{\psi}$) holds in low-level state $x$. *State-independent* predicates encode information that cannot be retrieved from the low-level state $x$, like object affordances (e.g. *graspable(?obj)*). We denote an evaluation of all predicates $\Psi$ on state $x$ as symbolic state $s = \{\underline{\psi} : c_{\underline{\psi}}(x) = \text{true}, \forall \psi \in \Psi\}$. Every operator $\omega \in \Omega$ consists of preconditions pre$(\omega)$ that define when $\omega$ can be executed, and effects eff$(\omega) = \langle EFF^+, EFF^- \rangle$ that define the add and delete effects of $\omega$ on the symbolic state $s$. We refer to an operator instantiated with objects $o_{1:n_\omega}$ for each variable $\mathcal{V}$ as action $a = \omega(o_{1:n_\omega})$. Additionally, eff$(a_{1:n})$ defines the state change resulting from applying plan $a_{1:n}$. We distinguish between skills $\pi$ and actions $a$: Actions are part of a high-level task plan, while skills are the low-level primitives the actor can execute on the continuous state $x$. An action $a$ is mapped to one or more skills $\pi$. Hierarchy levels are denoted with a superscript, where $D^{(0)}$ is the domain model at the upmost hierarchy level. A goal $g = \langle g^+, g^- \rangle$ is achieved if, for every $\underline{\psi} \in g^+$, $c_{\underline{\psi}}(x)$ is true, and for every $\underline{\psi} \in g^-$, $c_{\underline{\psi}}(x)$ is false (see Silver et al. (2023)).

Our objective is to learn domain models for long-horizon sequential tasks that require geometric and symbolic reasoning. We assume a deterministic environment $f$ and a list of all objects $\mathcal{O}$, which can be retrieved, for example, using segmentation (Kirillov et al., 2023). The environment state $x$ is fully observable. We assume a library $\Pi$ of skills with their names and parameter signatures. Executing a skill in the environment either transitions the environment into the next state or returns an exception that occurred during skill execution (e.g. for an invalid parameterization). Given a task specified by a natural language instruction $I$, we learn a hierarchical domain model that solves that and similar future tasks. Specifically, we construct operators, predicates and predicate classifiers.

## 4 METHODS

We propose LODGE, a method to learn explicit hierarchical PDDL domain models leveraging LLMs and environment grounding (see Fig. 1). We propose to **decompose** (see Sec. 4.1) domains hierarchically, which distributes the complexity of learning across hierarchy levels and enables faster planning for long-horizon tasks. We ensure the bidirectional **alignment** (see Sec. 4.2) between hierarchy levels, which is required to unify the subplans at different hierarchy levels to one valid hierarchical plan, avoiding infeasible plans due to sub-goal interactions. Plans are validated against the domain model, and executed in an environment. We **invent predicates and learn classifiers** (see Sec. 4.3) to estimate the symbolic state of the environment and compare it with the planner's state. A **global recovery module** (see Sec. 4.4) analyzes detected misalignments and suggests how to adapt the model to recover learning.

### 4.1 LEARNING HIERARCHICAL DOMAIN MODELS BY PLANNING

A central challenge in learning symbolic domain models is that initial models are incomplete and contain errors. We address this by incrementally constructing a hierarchical domain model, alternating between domain generation, planning, and environment grounding (see Fig. 1 and Alg. 1).

Given a natural language instruction $I$, skill library $\Pi$, a start state $x_0$, and an optional set of initial predicates $\Psi_{\text{init}}$, LODGE first induces a preliminary domain model $\mathcal{D}^{(0)}$ and goal state $g^{(0)}$:

$$\mathcal{D}^{(0)}, g^{(0)} = \text{LLM}_{\text{domain}}(I, c_\Psi(x_0), \Psi_{\text{init}}, \Pi).$$

The induced domain model $\mathcal{D}^{(0)}$ introduced new operators $\omega_i$ and predicates $\psi_j$, which are automatically classified as either *state-based* or *state-independent*. Only *state-based* predicates are considered for classifiers learning (see Sec. 4.3) and Motion Verification (see Sec. 4.4). The domain $\mathcal{D}^{(0)}$ and goal state $g^{(0)}$ are checked for syntax errors, and corrected with self-refinement (Howey et al., 2004).

**Abstraction through hierarchical operators** Instead of generating one operator for each pre-defined skill $\pi$, we explicitly instruct the LLM to define high-level operators. We later iteratively decompose these operators into lower-level domain models, forming a hierarchical domain model. This hierarchical approach provides two key benefits: (1) Multiple abstraction levels significantly reduce the domain model complexity by reducing the number of operators at one level and their

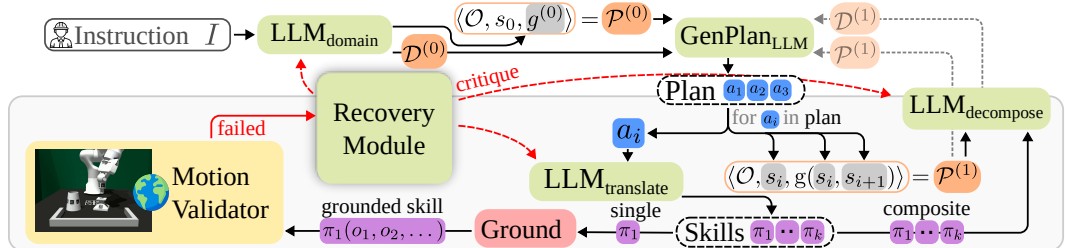

Figure 1: LODGE: Learning hierarchical domain models with environment grounding.

individual complexity, which also improve LLM accuracy for suggesting plans and reduces the exponential search time of classical planners. (2) Operators from a higher-level domain can operate on a coarse-grained symbolic state, while decomposition progressively refines state representation. Defining fine-grained predicates, e.g. the gripper position relative to an object, at the top level can be ambiguous and tracking their state complicates the operators (cf. Sec. 5.2).

**Plan generation under partial domain models**   LODGE uses domain $\mathcal{D}^{(0)}$ and problem $\mathcal{P}^{(0)} = \langle \mathcal{O}, s_0, g^{(0)} \rangle$ to generate a candidate plan $a_{1:n}^{(0)} = \text{GenPlan}_{\text{LLM}}(\mathcal{D}^{(0)}, \mathcal{P}^{(0)})$. At early stages, the classical planner often fails due to an incorrect domain. In such cases, LODGE reverts to an LLM-generated plan. While this plan may be invalid, it is required to find the misalignment between the domain model and environment during *Motion Verification* (see Sec. 4.4). In contrast to related work (Wong et al., 2023), we do not validate the plan correctness against the domain model, e.g. with VAL (Howey et al., 2004). VAL only detects unsatisfied preconditions of an action, but these can also be caused by missing effects of previous actions. Instead, the plan is verified during *Motion Verification*.

**Mapping to skills**   Given a plan candidate $a_{1:n}^{(0)}$, we iterate over every action $a_i^{(0)} = \omega(o_{1:n})$ to determine whether the operator $\omega$ must be decomposed or whether it can be realized with a single skill $\pi \in \Pi$ (Alg. 1). A translation model maps $\omega$ to a sequence of skills $\pi_{0:T-1} = \text{LLM}_{\text{translate}}(\omega, \Pi, \mathcal{O})$ parameterized by the operator's arguments. If this *lifted* skill sequence consists of a single skill, we mark $\omega$ as leaf node and start motion verification. We otherwise *decompose* the operator $\omega$ (Fig. 1).

**Decomposition**   The goal of decomposing an operator $\omega^{(0)}$ is to construct a lower-level domain $\mathcal{D}^{(1)}$. For example, the operator *grasp(bulb)* can be decomposed into *approach(bulb)*, *close_gripper(bulb)*, and *lift(bulb)*. The action $a_i^{(0)} = \omega^{(0)}(o_{1:n_\omega})$ in a sequence $a_{1:m}^{(0)}$ causes a state transition from $s_i$ to $s_{i+1}$, where $\text{pre}(\omega^{(0)})$ is a subset of $s_i$ and $\text{eff}(\omega^{(0)})$ describes the state change from $s_i$ to $s_{i+1}$ (Howey et al., 2004). To decompose $\omega^{(0)}$, we define a subproblem $P^{(1)}$ and subdomain $D^{(1)}$ with:

$$\mathcal{P}^{(1)} = \langle \mathcal{O}, s_i, \text{eff}(s_i, s_{i+1}) \rangle, \quad \text{where} \quad \text{eff}(s_i, s_{i+1}) = \langle s_{i+1} \setminus s_i, s_i \setminus s_{i+1} \rangle, \quad (1)$$

$$\mathcal{D}^{(1)} = \text{LLM}_{\text{decomp}}(\mathcal{P}^{(1)}, \Psi_{\text{init}}^{(1)}, \Pi, \omega^{(0)}, \pi_{0:T-1}),$$

where $\Psi_{\text{init}}^{(1)} := \Psi^{(0)}$ provides the initial predicate set. In contrast to $\text{LLM}_{\text{domain}}$, the decomposition model $\text{LLM}_{\text{decomp}}$ conditions on the high-level operator $\omega^{(0)}$ and problem $\mathcal{P}^{(1)}$, rather than on the natural language instruction $I$. We additionally provide the proposed skill sequence $\pi_{0:T-1}$ generated by $\text{LLM}_{\text{translate}}$ to align the decomposition with it. Importantly, once a decomposition $\mathcal{D}^{(1)}$ for $\omega^{(0)}$ is learned, it can be reused for any subsequent action $a_j = \omega^{(0)}(o'_{1:n_\omega})$ without additional LLM calls.

## 4.2   ALIGNING LEVELS OF HIERARCHICAL DOMAIN MODELS

Hierarchically decomposing the domain model presents two key challenges: (1) preserving knowledge across levels and (2) ensuring alignment between levels to ultimately produce a coherent hierarchical domain model and allow for hierarchical planning.

To preserve knowledge, we retain predicates $\Psi$ and objects $\mathcal{O}$ from one hierarchy level to all lower ones. Retaining the predicates for lower levels is crucial for aligning them with the environment during motion verification. However, lower-level predicates are not available at upper levels to

encapsulate state tracking. The decomposition of a high-level operator $\omega$ with predicates $\Psi^{(0)}$, and action $a = \omega(o_{1:n_\omega})$ with effects eff$(a)$ generates low-level operators, predicates $\Psi^{(1)} \supseteq \Psi^{(0)}$ and a subplan $a_{1:k}$ with joint effects eff$(a_{1:k})$. To keep $\omega$ aligned with the subplan, the effects eff$(a)$ must match the effects eff$(a_{1:k})$ on the upper-level predicates $\Psi^{(0)}$. It holds that eff$(a) \subseteq$ eff$(a_{1:k})$, as the goal state of the lower-level problem follows from the action's effects eff$(a)$. However, eff$(a_{1:k})$ can be a superset of eff$(a)$ such that more predicates in $\Psi^{(0)}$ change than initially assumed. For example, the high-level action *pick-up(apple)* has effect *grasps(apple)*. Decomposition leads to a joint effect [*grasps(apple)*, *door-open(drawer)*, *closed-gripper()*], eff$(a_{1:k})$, introducing two predicates not in eff$(a)$. We call these predicates eff$(a_{1:k}) \setminus$ eff$(a)$ *overshoots* and *side effects*: *Overshoots* operate solely on objects used in the action ($\forall o \in o_{1:n_\omega}$), while *side effects* operate on at least one object not used in the action; $\exists o \notin o_{1:n_\omega}$. In this example, *closed-gripper()* is an *overshoot* and *door-open(drawer)* a *side effect* of the decomposition.

These *misalignments* between hierarchy levels can break plan correctness at the upper level due to sub-goal interactions. After action $a$, the upper-level state reflects eff$(a)$, but executing the subplan $a_{1:k}$ changes the state by eff$(a_{1:k})$. Considering the example from above. If action *close-gripper()* follows next with precondition *not(closed-gripper)*, the plan appears valid at the upper level but fails after decomposition because the subplan's effects defines *closed-gripper* as true.

We detect misalignments by comparing eff$(a)$ with eff$(a_{1:k})$ on the upper-level predicates $\Psi^{(0)}$ after executing the subplan. If misalignments occur, LLM$_{\text{decomp}}$ realigns the operator $\omega$ of $a$ with its subplan $a_{1:k}$. Fixing *overshoots* is simpler than *side effects*, since *side effects* affects objects not used in $\omega$, which requires adding new variables to $\omega$, i.e. *drawer* in the example above. Smaller models like GPT 4o mini handle *overshoots* well but struggle with *side effects*.

## 4.3 Inventing Predicates and Learning Classifiers

The definition of predicates $\Psi$ can be non-trivial and can influence learning accurate domain models. Additionally, robotic environments do not expose symbolic states, requiring a classifier $c_\psi$ to ground predicates in the continuous environment state $x$. Apart from the initial predicates $\Psi_{\text{init}}$, LODGE invents predicates during domain learning as needed. We automatically learn predicate classifiers for newly invented *state-based* predicates when grounding is required.

We generate Python-based classifiers with an LLM as in Han et al. (2024), but prompt it to add hyperparameters $\boldsymbol{\theta}$ with plausible default values $\boldsymbol{\theta}^{\text{default}}$ as arguments to the function, e.g. 'def holding(obj: Part, *xy_tolerance: float = 0.05*)'. The Python code has access to the perceived object poses and the robot end-effector state (see App. E). Compared to LLMs or VLMs, Python-based classifiers are explicit, interpretable, and significantly more efficient. Additionally, relying on the perception of the environment instead of LLMs avoids hallucinations. However, the generated code can contain mistakes, i.e. syntax or logical errors. Additionally, most classifiers depend on hyperparameters (see above or Fig. 2, Han et al. (2024)). Robust values for these hyperparameters vary with the system and environment, e.g. the perception noise. To address these challenges, we propose (1) a method to collect pseudo-labeled interactions and (2) algorithms to optimize the Python-based classifiers on that data.

**Data Collection** We record all environment transitions $t_i = (x_i, \pi_i, x_i') \in \mathcal{T}$ during Motion Verification (see Sec. 4.4), consisting of state $x_i$, skill $\pi_i$, and next state $x_i'$. For all next states $x_i'$ in the transitions, we predict pseudo-labels $s_i'$ similar to Athalye et al. (2025) (see App. D). This dataset $\mathcal{D} = \{(x_i, \pi_i, s_i)\}_{i=1}^n$ is used to evaluate and refine the predicate classifiers. Given a constructed Python classifier $c_\psi$ and dataset $\mathcal{D}$, we evaluate $c_\psi$ by computing the F1 score $\mathcal{F}_\psi$ on $\mathcal{D}$ for every ground atom $\psi$. We refine the predicate classifier based on the lowest F1 score of any ground atom. On a F1 score below $\tau_{\text{hp}}$, we refine the classifier's hyperparameters $\theta$ for a better evaluation metric via *hyperparameter optimization*. When the F1 score is below $\tau_{\text{llm}}$, we improve the code of the classifier via *LLM self-refinement*. $\tau_{\text{llm}}$ is generally lower than $\tau_{\text{hp}}$, suggesting an error in the code of the classifier. We empirically used $\tau_{\text{hp}} = 0.9$ and $\tau_{\text{llm}} = 0.6$ for all our experiments.

**Hyperparameter Optimization** We optimize the LLM-given default hyperparameters $\boldsymbol{\theta}^{\text{default}}$ via gradient-free random search evaluated on the average F1 score $\mathcal{F}(\boldsymbol{\theta}) = \langle \mathcal{F}_\psi(\boldsymbol{\theta}) \rangle_\psi$. Among all samples that achieve the best F1 score $\mathcal{F}$, we first identify the most robust candidates and then select

the one closest to $\boldsymbol{\theta}^{\mathrm{default}}$. We evaluate the robustness $R(\boldsymbol{\theta})$ of $\boldsymbol{\theta}$ as the minimal relative change in any hyperparameter $k$ that changes the F1 score $\mathcal{F}(\boldsymbol{\theta})$,

$$R(\boldsymbol{\theta}) = \min_{\boldsymbol{\theta}': f(\boldsymbol{\theta}') \neq f(\boldsymbol{\theta})} \quad \min_{k} \frac{|\theta'_k - \theta_k|}{|\theta_k^{\mathrm{default}}|}.$$

**LLM Self-Refinement**  We invoke the LLM to refine the Python classifier. For this, we sample up to three samples from $\mathcal{D}$ on which the Python classifier yields a different classification than the pseudo-label and prompt the LLM with the current Python classifier and the samples to either refine the code, or leave it as is if it believes the VLM labels are incorrect. Every sample in the prompt is stated by the 6D poses of robot and objects, the classification results of the Python classifier and VLM, and the classification result of the Python classifiers used in the Python code (see App. D).

### 4.4 VERIFYING PLANS, RECOVERING FROM MODEL ERRORS, AND ADAPTING LEARNING

To evaluate domain model correctness, we execute every planned skill in simulation and compare the planner's symbolic state change with the observed state change in simulation. A misalignment indicates an error in the learned domain model that we address with a *global recovery module*.

**Motion Verification**  Given a leaf action $a = \omega(o_{1:n_\omega})$, the mapped skill $\pi$, and the current state of the environment $x_t$, we first verify that the preconditions of $\omega$ hold in the environment, such that $\mathrm{pre}(\omega) \subseteq c_{\underline{\Psi}}(x_t)$. We then execute $\pi$ in the environment, returning the next state $x_{t+1} = f(x_t, \pi(o_{1:n_\pi}))$. We lastly verify that the effects $\mathrm{eff}(\omega)$ equal those observed in the environment, such that $\mathrm{eff}(\omega) = \mathrm{eff}(c_{\underline{\Psi}}(x_i), c_{\underline{\Psi}}(x_{i+1}))$ (see Eq. 1). We verify that *state-based* predicates hold in the simulation. *State-independent* predicates define relations that do not change during planning and cannot be grounded from the low-level environment state, e.g. *openable*. We have to rely on the LLM to detect incorrectly defined *state-independent* predicates, as we can not verify them in simulation.

**Global Recovery Module**  A failed verification for an action $a_i$ in a plan $a_{1:k}$ can be caused by violated preconditions, misaligned effects, or an incorrect mapping to a predefined skill. We propose a *global recovery module* $\mathrm{LLM}_{\mathrm{reasoner}}$ that analyses the occurred verification failure to determine *how to adapt the domain model*. $\mathrm{LLM}_{\mathrm{reasoner}}$ oversees the past model learning, with access to the history of previous LLM interactions, and a summary of the motion verification failure, either showing the state misalignment, or the occurred exception raised while executing the skill. $\mathrm{LLM}_{\mathrm{reasoner}}$ then determines the cause for the misalignment in the past model learning and suggests *where* to adapt the domain model and *how* to continue learning by outputing the operator $\omega$ that should be adapted. Once decided, we retract to the affected level and re-prompt the related module to incorporate the fix.

## 5 EVALUATION

We evaluate LODGE on three domains to answer (1) How well does LODGE learn operators with little predefined information and few environment interactions (Sec. 5.1)? (2) How well can LODGE learn entire domains (predicates, predicate classifiers, and operators) (Sec. 5.2)? (3) Can LODGE optimize predicate classifiers from pseudo-labels (Sec. 5.2)?

### 5.1 LEARNING OPERATORS FOR IPC DOMAINS

We evaluate operator learning with LODGE on the *household* and *logistics* IPC domains. Since these domains operate on discrete state spaces, we do not learn predicate classifiers but ground the planner's state in the environment. Additionally, we focus on learning operators and assume the *state-based* predicates are known. LODGE learns operators for the first task of a domain and then refines it for each following task. We limit environment interactions to 10 and re-planning iterations (e.g. to fix syntax errors) to 20 per task to demonstrate LODGE's efficiency in learning operators with few interactions. We use the instructions and domain descriptions from Guan et al. (2023), but leave out their skill annotations, as they give a detailed description about the skills and are generally unavailable (see App. G for a exemplary skill).

We compare LODGE with Guan et al. (2023) (without providing human feedback) and Mahdavi et al. (2024). We additionally evaluated Wong et al. (2023)'s method for domain learning, but it

| Env | Model | Guan | Mah. w/ ann. | Mah. w/o ann. | LODGE (ours) |
|---|---|---|---|---|---|
| Logistics | Llama 4 Scout | 0.09 / 0.05 | 0.97 / 0.95 | 0.70 / 0.37 | 0.92 / 0.87 |
| | Llama 4 Mav. | 0.08 / 0.05 | 1.00 / 1.00 | 1.00 / 1.00 | 1.00 / 1.00 |
| | GPT 4.1 mini | 0.90 / 0.68 | 1.00 / 1.00 | 0.99 / 0.98 | 1.00 / 1.00 |
| Household | Llama 4 Mav. | 0.03 / 0.00 | 0.38 / 0.00 | 0.07 / 0.00 | 0.57 / 0.25 |
| | GPT 4.1 mini | 0.06 / 0.00 | 0.42 / 0.00 | 0.20 / 0.03 | 0.69 / 0.44 |
| FB Lamp | | 0.26 / 0.06 | 0.30 / 0.25 | 0.23 / 0.17 | 0.78 / 0.81 |
| FB Round Table[†] | GPT 4.1 mini | 0.21 / 0.15 | 0.32 / 0.15 | 0.11 / 0.00 | 0.71 / 1.00 |
| FB Square Table | | 0.16 / 0.00 | 0.20 / 0.00 | 0.20 / 0.00 | 0.53 / 0.20 |
| FB Stool & Desk[‡] | | 0.11 / 0.00 | 0.24 / 0.00 | 0.16 / 0.00 | 0.52 / 0.14 |

Table 1: Avg@3 (Exploration Walk/Tasks solved) for six domains. [†]LODGE reuses and refines the domain model learned from FB Lamp. [‡]LODGE reuses and generalizes the domain model learned from FB Square Table.

did not succeed in generating domain models for any of the two IPC domains (see Appendix F). For each learned domain model, we evaluate task success and estimate similarity to a predefined domain model using an adapted Exploration Walk (EW) metric (Mahdavi et al., 2024) (see Appendix H). Each method is evaluated three times per domain, and results are averaged. The EW metric is defined as the harmonic mean of the two percentages that indicate how many plans randomly sampled from one domain are executable in the other domain. A value of 1.0 indicates that both domains are identical. We compare all methods on GPT 4.1 mini and Llama 4 Maverick, and additionally evaluate the smaller Llama 4 Scout on the *logistics* domain.

Table 1 evaluates the two IPC domains. While all methods perform well on *logistics* with GPT 4.1 mini, LODGE remains robust with weaker LLMs like Llama 4 Scout, whereas Mahdavi et al. (2024) struggles without skill annotations. Moreover, for the *household* domain, LODGE produces domain models closer to the predefined domain model, solving up to 44 % of the tasks, while the domains of other approaches rarely yield successful plans. Guan et al. (2023) and Mahdavi et al. (2024) by default use skill annotations for domain learning. To assess the relevance of these annotations to their methods, we evaluate Mahdavi et al. (2024) without passing the annotations (see Table 1 under 'Mah. w/o ann.'). As Guan et al. (2023) solely rely on skill annotations, we omit them from the analysis. Leaving out the annotations for Mahdavi et al. (2024) decreases the domain model correctness for both IPC domains significantly. Especially for the more complex *household* domain, Mahdavi et al. (2024) relies on the annotations and struggles to generate an accurate domain model without them.

## 5.2 LEARNING DOMAIN MODELS FOR ROBOTICS DOMAINS

Robotics domains differ from IPC domains by being continuous with generally executable but uncertain skills. In this experiment, we learn entire domain models with LODGE, comprising predicates, predicate classifiers, and operators. LODGE only receives the goal predicate '*assembled(?obj1, ?obj2)*' in the initial predicates $\Psi_{\text{init}}$, whose classifier indicates whether two parts have been assembled, as defined by the FurnitureBench benchmark (Heo et al., 2023). Since the planner's state and the continuous state of the environment are independent, we additionally construct and refine predicate classifiers for newly invented *state-based* predicates (see Sec. 4.3). We evaluate domain model learning as in Section 5.1 and compare our algorithm to learn predicate classifiers to related methods.

**Domain Model Evaluation** Table 1 evaluates the learned domain model of the FurnitureBench domain and task success in assembling the *lamp* and *square-table*. We additionally evaluate generalizing both domains to similar tasks, namely *round-table* for *lamp* and *desk* and *stool* for *square-table*. The *square-table* domain differs from the *lamp* domain because it contains parts that have multiple threaded holes for assembly. While Mahdavi et al. (2024) significantly outperforms Guan et al. (2023) in IPC domains, it struggles in FurnitureBench. The main cause lies in

| Method | F1 Score |
|---|---|
| InterPreT (Py) | 0.71 |
| Pix2Pred (VLM) | 0.84 |
| LODGE (ours) | 1.00 |

Table 2: Predicate Grounding

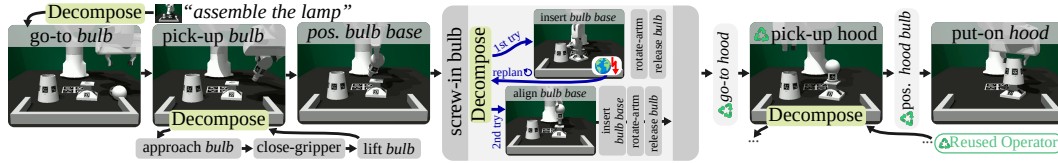

Figure 2: The hierarchical planning of assembling the *lamp* with LODGE for the FurnitureBench environment, including decomposition of *pick-up(bulb)* and re-planning within *screw-in(bulb)*.

the EW metric, which, although smooth, decreases exponentially with the number of incorrect terms. As a result, correcting individual errors for large (*household*) or complex (*FurnitureBench*) domains often has no effect on the score. Moreover, the language feedback in Mahdavi et al. (2024) is restricted to syntax errors and does not identify which operators or plans are problematic, which makes improvements difficult. LODGE addresses these limitations by proposing a targeted adaptation of operators based on the predicate classifiers' feedback (see App. I). As a result, LODGE assembles the lamp in two of three seeds fully and achieves an average of 81 % of the lamp assembly.

Figure 2 depicts the hierarchical domain learning of assembling *bulb*, *base* and *hood* to a *lamp*. The decomposition significantly reduces the plan length at the topmost level. LODGE invents predicates at the top level to capture high-level object relations, like *aligned* or *holding*. During decomposition, LODGE invents fine-grained predicates to track the state as needed, i.e. *gripper-around-part* for the decomposition of *pick-up* or *touching* for the decomposition of *screw-in*. Additionally, LODGE reuses the decomposed domain model of *pick-up(bulb)* for *pick-up(hood)*.

**Predicate Classifier Evaluation** While learning the hierarchical domain model, LODGE invents predicates and learns classifiers. We compare our method of learning classifiers to InterPreT (Han et al., 2024) and Pix2Pred (Athalye et al., 2025). Table 2 shows the evaluation of all methods on the states observed while assembling the *lamp*, evaluated on the F1 score $\mathcal{F}$ metric of Section 4.3. The VLM in Athalye et al. (2025) generates more accurate predicates than writing a Python classifier. We found that the Python classifier for some predicates yields a significantly lower F1 score than on others. This is either caused by an incorrect Python implementation (e.g. too restrictive checks such as comparing part rotations), coding errors, or unsuitable tolerances (e.g. *on-table* uses a 1 mm distance tolerance, while perception noise is on the order of 1 cm).

| Planner Type | # It. | LOG | HH |
|---|---|---|---|
| LLM Planner Ann.=yes | 0 | 1 of 21 | 3 of 24 |
| | 10 | 18 of 21 | 11 of 24 |
| LLM Planner Ann.=no | 0 | 1 of 21 | 1 of 24 |
| | 10 | 13 of 21 | 6 of 24 |
| | 20 | 13 of 21 | 13 of 24 |
| LODGE (ours) | 10 | 21 of 21 | 11 of 24 |

Table 3: Task success in Logistics (LOG) and Household (HH) domains, comparing domain learning with refining plans via environment interactions (#It.) and skill annotations (Ann.).

In comparison, our method learns accurate Python classifiers by refining them with pseudo-labels.

**Real-World Evaluation** We further evaluate LODGE on a real environment to test both the transfer of a domain model learned in simulation and the ability to learn hierarchical domain models directly from real interactions. We captured object poses with OptiTrack and recorded images with a static camera, which did not noticeably impact VLM accuracy. In comparison to simulation, we manually reset the environment to the task start when necessary. We refer to Ablation B for a detailed description of the setup. Despite increased environment variance and object tracking noise compared to the simulation, the domain model learned on the real system achieved substantially higher accuracy than baseline methods, successfully assembling the lamp by 38 %.

### 5.3 ABLATION

**How does domain learning compare to plan refinement?** Instead of learning domain models and thereby learning *how to plan*, we can adapt the plan directly on the environment. Table 3 compares the plan success of LODGE against an LLM planner (Valmeekam et al., 2023) that directly refines the plan, with and without access to skill annotations. LODGE solves

more tasks with the same amount of environment interactions (# Iters.) while jointly learning a domain model, which prevents future environment interactions to solve similar tasks.

**How does LODGE compare to direct operator learning?** Having invented predicates and predicate classifiers, we can construct operators with classical algorithms (Silver et al., 2021b) on the interaction dataset $\mathcal{D}$ instead of using LODGE's LLM-based method for operator learning. Therefore, we evaluate how LODGE compares to classical methods for operator learning on the same data. Table 4 compares the approaches of Guan et al. (2023), Mahdavi et al. (2024) and LODGE with the cluster-and-search (C&S) algorithm proposed by Silver et al. (2021b). We evaluate C&S on the same interactions LODGE sees

| Method | Domain Eval. |
|---|---|
| Guan | 0.26 / 0.06 |
| Mahdavi w/o Annot. | 0.23 / 0.17 |
| LODGE (ours) | 0.71 / 0.78 |
| C&S w. Py | 0.23 / 0.11 |
| C&S w. VLM | 0.17 / 0.00 |

Table 4: Fitting operators directly on the environment interactions

during domain learning. *C&S w. Py* uses labels generated by the correct predicate classifiers, while *C&S w. VLM* uses Pix2Pred (Athalye et al., 2025) for labeling, which also contains misclassifications. The Table shows that operator learning with classical algorithms on limited amount of data reaches a low performance. Our method of combining LLMs and predicate classifiers for grounding significantly outperforms directly inferring operators for a small dataset (20 transitions).

**Does predicate grounding improve domain models?** We evaluate LODGE without learning predicate classifiers in the FurnitureBench *lamp* assembly domain. Table 5 shows that not grounding the planners' state in the environment significantly impacts domain model correctness and planning success, underlining the importance of grounding domains in the environment.

| Grounding | EW / Succ. |
|---|---|
| w/ | 0.78 / 0.71 |
| w/o | 0.38 / 0.14 |

Table 5: Importance of predicate grounding for LODGE on *FurnitureBench*

**Is LODGE robust to the skill set size?** LODGE constructs operators only for skills needed to solve the current task, and hierarchical abstraction reduces the number of operators at each level. We evaluate the effect of skill set size, by increasing the number of skills from 22 to 72 in the *household* domain. Despite the threefold increase in available skills, the number of learned operators did not increase, and the use of input and output token increased only slightly by 44% and 25%, respectively.

## 6 CONCLUSION

We propose LODGE, a framework that uses LLMs and environment grounding to learn hierarchical domain models by iteratively refining an abstract domain into subdomains. We propose to learn classifiers for newly invented predicates and refine them on pseudo-labeled environment interactions. A novel *global recovery module* analyzes misalignments between the domain model and environment by evaluating the classifiers and guides realigning the domain with the environment. Evaluation on three domains shows that LODGE learns more accurate domain models that yield higher task success than existing methods. On the robotics domain, we demonstrate that LODGE can refine domain models from learned classifiers for continuous environments that are not tailored to formal languages.

While LODGE shows strong performance on hierarchical domain learning, several directions remain for future work. First, domain decomposition requires aligning hierarchy levels, which is non-trivial due to goal overshoots and side effects. Future work could investigate on more robust approaches for smaller LLMs. Second, our robotics evaluation is limited to the FurnitureBench domain, as no benchmarks exist for domain learning on continuous environments. Existing robotics benchmarks either target RL or operate on discrete spaces (Silver and Chitnis, 2020). Third, constructing large domain models with LODGE from scratch is slow. More sophisticated decomposition could improve learning speed, e.g. by sharing operators across hierarchy levels. Finally, combining LODGE with efficient, constraint-based path planners could realize highly flexible Task and Motion Planning (TAMP) systems (Silver et al., 2021b), which requires further research into representations bridging PDDL-style planners, constraint-based motion planning and LLM reasoning.

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

---

**Algorithm 1** Recursive Operator Decomposition in LODGE

---

**Require:** Instruction $I$, initial predicates $\Psi_{\text{init}}$, skill library $\Pi$
**Ensure:** Hierarchical domain model $\langle \mathcal{D}^{(0)}, \ldots, \mathcal{D}^{(d)} \rangle$
1: **function** TRAVERSE($\mathcal{D}^{(i)}, \mathcal{P}^{(i)}, \Pi$)
2:        Learn classifiers $c_{\psi_{\text{new}}}$ for new *state-based* predicates $\psi_{\text{new}}$ in $\mathcal{D}^{(i)}$
3:        Generate plan $a^{(i)}_{1:n} = \text{GenPlan}_{\text{LLM}}(\mathcal{D}^{(i)}, \mathcal{P}^{(i)})$
4:        **for** each action $a^{(i)}_j$ in $a^{(i)}_{1:n}$ **do**
5:            Suggest skills $\pi_{0:T-1} = \text{LLM}_{\text{translate}}(a^{(i)}_j, \Pi, \mathcal{O})$
6:            **if** length($\pi_{0:T-1}$) = 1 **then**
7:                Motion Validation of $\omega^{(i)}_j$              ▷ leaf operator
8:            **else**
9:                Construct $\mathcal{P}^{(i+1)} = \langle \mathcal{O}, s_j, \text{eff}(s_j, s_{j+1}) \rangle$
10:             Generate $\mathcal{D}^{(i+1)} = \text{LLM}_{\text{decomp}}(\mathcal{P}^{(i+1)}, \Psi_{\text{init}}, \Pi, \omega^{(i)}_j, \pi_{0:T-1})$
11:             TRAVERSE($\mathcal{D}^{(i+1)}, \mathcal{P}^{(i+1)}, \Pi$)       ▷ recursive decomposition
12:            **end if**
13:        **end for**
14: **end function**
15: Initialize $\mathcal{D}^{(0)}, g^{(0)} = \text{LLM}_{\text{domain}}(I, c_{\Psi}(x_0), \Psi_{\text{init}}, \Pi)$
16: TRAVERSE($\mathcal{D}^{(0)}, \mathcal{P}^{(0)}, \Pi$)

---

## A   HIERARCHICAL DOMAIN LEARNING

Algorithm 1 shows the learning of a hierarchical domain model.

### A.1   ABLATING LLM USAGE

In the following, we ablate the token usage of the LLM of all methods. While Guan et al. (Guan et al., 2023) and Mahdavi et al. (Mahdavi et al., 2024) learn one domain, we learn task-centric domain models that are extended for every new task. For a fair comparison of the token consumption, we generate the same domain model with all methods. The domain model consists of five operators of the household domain and the predicates used by these operators. Table 6 shows the usage needed to generate the domain model per method, averaged over three seeds: LODGE adapts the initially

| Method | Input Tokens | Output Tokens | Number of Calls |
|---|---|---|---|
| LODGE | 120k | 9k | 18 |
| Guan | 276k | 40k | 144 |
| Mahdavi w/o annot | 138k | 163k | 223 |
| Mahdavi w annot | 157k | 193k | 224 |

Table 6: Token consumption and LLM call statistics for domain generation, averaged over three seeds.

generate domain model in a targeted manner. This significantly reduces output token usage. In contrast, Mahdavi et al. (Mahdavi et al., 2024) generate five problem candidates and ten domain candidates, which explains the higher consumption of output tokens.

### A.2   FLAT VS HIERARCHICAL DOMAIN MODEL LEARNING

We propose to learn hierarchical domain models with LODGE, which simplifies domains on every level and distributes the complexity of learning the domain model across multiple levels. Table 7

compares learning flat domain models with learning hierarchical domain models with LODGE. In both cases, we learn the domain model for the FurnitureBench lamp task (Heo et al., 2023) from three seeds and average the results. We evaluate both domains as in Section 5.2. Table 7 clearly indicates

| Configuration | Exploration Walk | Tasks Solved |
|---|---|---|
| Flat Domain Model | 0.30 | 0.13 |
| Hierarchical Domain Model | 0.78 | 0.81 |

Table 7: Comparison of flat vs. hierarchical domain learning on FurnitureBench and LAMP, averaged over 3 seeds.

that hierarchical domain models promote learning. In learning a flat domain model, we found two causes complicating learning: (1) A constructed operator must always be matched to exactly one predefined skill, which requires significant operator refinements. With hierarchical domain models, the LLM can generate more abstract operators that do not have to align with exactly one predefined skill. (2) The entire predicate state must be tracked on one domain level. For instance for the grasping of parts, this requires multiple predicates, e.g. approaching an object, surrounding it, and finally grasping it. All operators in a flat domain model have to track these fine-grained predicates. In hierarchical domain models however, this fine-grained state is enclosed in a subdomain, such that operators on upper levels do not have to track these predicates.

# B  REAL WORLD EXPERIMENTS

We evaluate LODGE in a real world environment to answer two questions: (1) How reliably can a domain model learned in simulation be used to plan skill sequences in the real world. (2) Can LODGE learn hierarchical domain models directly from real-world interaction. For both experiments, we use a real world environment to assemble the FurnitureBench lamp (see Figure 3). To estimate the object localizations, we use OptiTrack (Opt) with 4-5 markers attached to each object. Additionally, we implement the same skill library used for the simulation-based experiments for the real world with libfranka. Lastly, we mount a static RealSense D405 camera to capture RGB images of the scene for the VLM groundings.

**How reliably can a domain model learned in simulation be used to plan skill sequences in the real world?**   To answer this question, we use the domain models learned with LODGE in the simulated environment from Section 5.2. We plan the lamp assembly five times and execute the skill sequence on the real robot. In three of the five executions, the robot fully assembled the lamp. In one execution, the robot failed in inserting the bulb in the lamp's base due to a too inaccurate object localization. One of the five plans was invalid, causing the assembled bulb and base to be thrown over by the robot while approaching the hood.

**Can LODGE learn hierarchical domain models directly from real-world interaction?**   LODGE learns the hierarchical domain model by evaluating plan candidates in an environment and refining the domain model based on the environment feedback. In this experiment, we learn the domain model for the lamp assembly directly on the real robot. As in the simulation experiments, we limit the number of plan candidates evaluated to ten, keeping the amount of plan trials low. We use OptiTrack to capture object poses, though hand-eye calibration errors and the Franka controller's imprecision introduce noise of approximately 1 cm. Additionally, pseudo-labels are used as in simulation, while we observed that the real-world images do not impact VLM accuracy. Resetting the environment to a specific state in the middle of a plan sequence is challenging on the real environment. Instead, we manually reset the environment to the start of the task after recovering from a *misalignment* and continue learning from there. Aside from these adaptations, the learning procedure follows the same setup as Section 5.2. Table 8 compares the domain model learned with LODGE on the real system to the methods introduced in Section 5.2. The reduced correctness of the real-world domain model, compared to the one learned with LODGE in simulation, can be attributed to increased variance and noise. For instance, in simulation, resetting the environment always restores it to the exact same

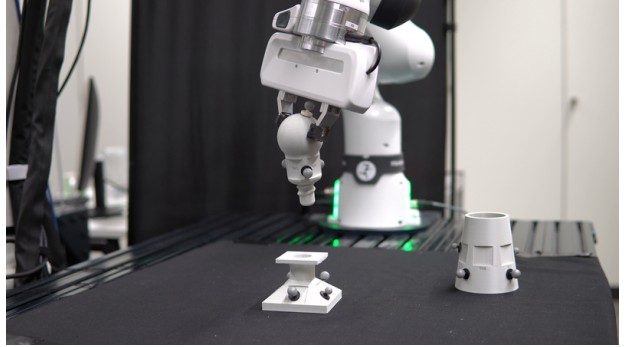

Figure 3: Real world setup for assembling the lamp of the FurnitureBench benchmark (Heo et al., 2023).

| Method | EW/Succ. |
|---|---|
| Guan | 0.26 / 0.06 |
| Mah. w/ ann. | 0.30 / 0.25 |
| Mah. w/o ann. | 0.23 / 0.17 |
| LODGE (sim) | 0.78 / 0.81 |
| LODGE (real) | 0.48 / 0.38 |

Table 8: (Exploration Walk/Tasks solved) Comparison of domain-model learning for the FurnitureBench lamp assembly on the real robot using LODGE versus all methods in simulation.

state. On the real system, we manually reset the objects to their start configuration, but their poses inevitably differ with each reset. Despite this, the domain model learned on the real system remains more accurate than the baselines, even when skill annotations are included.

## C  ERROR REASONER

The central error reasoner can invoke corrections at any hierarchy level. We alternatively tested retracting one hierarchy level at a time to self-verify each previously instructed LLM given the observed simulation error. However, the LLMs tend to correct themselves even if they cause for the misalignment lies somewhere else. Additionally, a level-by-level retracting is more computationally expensive with increasing number of hierarchy levels, making it unprobable to retract multiple layers, while the central reasoner can skip any hierarchy levels ad trigger re-planning.

To further enhance coherence, we introduce an in-place modification mechanism for LLM messages when traversing within a hierarchy level. This mechanism summarizes information or results from lower levels and injects them as a single, refined message, as if the LLM had directly produced the correct output without enlargening the chat history. For example, during decomposition, the LLM initially proposes a domain, which is iteratively refined to correct syntax, semantics, or planning infeasibility. The final, corrected domain is then re-injected into the original response, keeping the chat history compact and concealing intermediate errors.

## D  VLM GROUNDING

We obtain $s_i'$ from a VLM (as in Athalye et al. (2025)) given the transition $t_i$ and an image of the scene at state $x_i'$. The start state $x_0$ is labeled directly from $x_0$ and the image.

To reduce the number of samples to ground with the VLM, we compare the similarity of a new state with previously seen states. Only significantly new states are grounded by the VLM. A state that is similar to an existing state uses the same VLM grounding. The intuition for this is that the image of similar states are very similar. To estimate similarity, we compute the equivalence class for every sample by

$$\text{neigh}(\pi, x) = |\{(x_j, \pi_j, s_j) \in \mathcal{D} : \pi_j = \pi \land \|x_j - x\| < \tau\}|.$$

Only the first sample of this equivalence class is grounded by the VLM. All other samples in that class copy that grounding. To ground the states with the VLM, we use the same prompts as Athalye et al. (2025). We additionally pass known perception information and predicate descriptions to the VLM, e.g.

```
Pose estimation of all objects and the robot gripper (3D poses are in meters):
arm
- gripper_center: [0.567, 0.055, 0.124]
- gripper_closed: False
table
- surface_z: -0.016
```

```
lamp_base
- center: [0.455, 0.050, 0.017]
- orientation: [3.140, -0.000, -1.380]
lamp_bulb
- center: [0.433, 0.195, 0.015]
- orientation: [-3.140, 1.420, 0.040]
lamp_hood
- center: [0.459, -0.110, 0.035]
- orientation: [3.140, -0.000, 1.540]
Predicates:
# Grounding assembled: Evaluates whether obj1 has been assembled with obj2 the way they
    should be, e.g. when obj1 has been screwed into obj2, or obj1 has been put on top of
    obj2.
assembled(lamp_base, lamp_bulb)
...
```

## E  PYTHON PREDICATE CLASSIFIER

All Python classifiers can use following perception API.

```python
from dataclasses import dataclass
from typing import Tuple
import math

@dataclass
class Part:
    bounding_box: Tuple[float, float, float, float, float, float]  # (x_min, y_min, z_min,
     x_max, y_max, z_max)
    center: Tuple[float, float, float]  # (x, y, z)
    orientation: Tuple[float, float, float]  # (roll, pitch, yaw) in radian

@dataclass
class Table:
    surface_z: float

@dataclass
class Robot:
    gripper_center: Tuple[float, float, float]  # (x, y, z) of midpoint between jaws
    gripper_closed: bool

def angle_diff(a: float, b: float):
    """Compute the minimal difference between two angles in radians."""
    return ((a - b) + math.pi) % (2 * math.pi) - math.pi

def get_robot() -> Robot: ...
def get_table() -> Table: ...
def get_all_parts() -> List[Part]: ...
```

Additionally, a predicate classifier can call other already invented classifiers.

### E.1  GENERATE PYTHON CLASSIFIER

The prompt to the LLM to generate a predicate classifier for *aligned(?obj1, ?obj2)* is

```
Domain Knowledge:
{domain description}

Given Python code:
```python
{perception api}

def holding(r: Robot, p: Part) -> bool:
    """The robot is currently holding the part with its gripper.
    Returns: True if the predicate holds, False otherwise.""" ...
def on_table(p: Part, t: Table) -> bool:
    """The part is resting on the table.
    Returns: True if the predicate holds, False otherwise.""" ...
```

```python
def aligned(p1: Part, p2: Part) -> bool:
    """The held part is aligned with the fixed part for assembly (xy position and roll/pitch
      orientation aligned).
    Returns: True if the predicate holds, False otherwise.""" ...
```
```

```
Predicate: (aligned ?p1 - part ?p2 - part): The held part is aligned with the fixed part for
     assembly (xy position and roll/pitch orientation aligned).

Can you add the python body that implements the grounding of this predicate?
You must define hyperparameters or constants as kwargs with default values.
Where possible, use the already defined python functions to reduce code duplication.
Your response should contain two sections
[START OUTLINE]
# Requirements to Check
[list all requirements the function must validate. Only include checks that can be performed
     with the provided context (skip/ignore others).]
# Predicate Grounding
[insert the python code enclosed in '''python '''. If you think the predicate cannot be
     grounded reliably with the provided context, return 'none' and no python code. Don't
     define methods you do not implement or rely on information not provided.]
# Grounder Description
[insert a description what the grounder function tests for. It should be short but complete
     .]
[END OUTLINE]
```

The prompt contains python functions from already invented classifiers *holding*, *on_table*, and *aligned*. Also, it can be seen that the signature of the Python functions mimics the predicate signature.

The LLM then defines the function body of the predicate classifier in its response and adds required hyperparameters as additional keyword arguments to the function signature.

```
[START OUTLINE]
# Requirements to Check
- The xy positions of p1 and p2 are within a given positional tolerance ('pos_tol').
- The roll angles of p1 and p2 differ by no more than a given angular tolerance ('angle_tol')
     .
- The pitch angles of p1 and p2 differ by no more than a given angular tolerance ('angle_tol
     ').
- The yaw difference is ignored.

# Predicate Grounding
'''python
def aligned(p1: Part, p2: Part, pos_tol: float = 0.01, angle_tol: float = 0.1) -> bool:
    """The held part is aligned with the fixed part for assembly (xy position and roll/pitch
      orientation aligned).

    Args:
        p1: The held part.
        p2: The fixed part.
        pos_tol: Positional tolerance in meters for xy alignment.
        angle_tol: Angular tolerance in radians for roll and pitch alignment.

    Returns:
        True if aligned, False otherwise.
    """
    # Check xy position closeness
    dx = p1.center[0] - p2.center[0]
    dy = p1.center[1] - p2.center[1]
    pos_close = (dx * dx + dy * dy) <= (pos_tol * pos_tol)

    # Check roll difference
    roll_diff = abs(angle_diff(p1.orientation[0], p2.orientation[0]))
    # Check pitch difference
    pitch_diff = abs(angle_diff(p1.orientation[1], p2.orientation[1]))

    return pos_close and roll_diff <= angle_tol and pitch_diff <= angle_tol
'''
# Grounder Description
The grounder checks whether two parts are aligned by verifying that their xy positions are
     within a specified positional tolerance and that their roll and pitch angles differ no
     more than a given angular tolerance. The yaw angle difference is ignored since it is not
      required for alignment.
[END OUTLINE]
```

## E.2  REFINE PYTHON CLASSIFIER

The prompt to refine Python classifiers contains the current implementation, as well as up to three samples from the interaction dataset $\mathcal{D}$. It additionally provides the perception information for every samples, as well as the classifier's output and the pseudo-label.

```
{domain description}
```

```
{code api}

{current predicate classifier}

This is a function that grounds the PDDL predicate. However, we evaluated it against a
    dataset labeled by an VLM and found following errors:
1.
- VLM predicates: (gripper_around lamp_bulb)
- Grounder predicates: (not (gripper_around lamp_bulb))
- Referenced Predicate Evals:

- Variables:
    -arm: Robot(gripper_center=[0.456, 0.183, 0.014], gripper_closed=False)
    -table: Table(surface_z=-0.016)
    -lamp_base: Part(bounding_box=[0.405, 0.002, -0.015, 0.505, 0.101, 0.048], center=[0.455,
     0.052, 0.017], orientation=[3.14, -0.0, -1.38])
    -lamp_bulb: Part(bounding_box=[0.37, 0.149, -0.015, 0.49, 0.209, 0.045], center=[0.43,
    0.179, 0.015], orientation=[-1.75, 0.54, 1.5])
    -lamp_hood: Part(bounding_box=[0.415, -0.154, -0.015, 0.503, -0.066, 0.085], center
    =[0.459, -0.11, 0.035], orientation=[-3.14, -0.0, 1.54])

    Your response should contain three sections
[START OUTLINE]
# Error Analysis
[evaluate the errors by tracing the values and result in the code.]
# Suggested Fixes
[list all checks in the code that contributed to the error. For every of them, verify they
    are required given information about the predicate. Consider also new checks that should
     be implemented.]
# Fixed Code
[insert your suggested changes and output the updated Python code. The fixes should correct
    the function to fix as many errors as possible while not violating the predicate
    definition and description. Do not add methods with a placeholder body or `
    NotImplementedError`.]
# Grounder Description
[insert a description what the grounder function tests for. It should be short but complete
    .]
[END OUTLINE]
```

Based on that, the LLM analyzes the cause for the different classifications of VLM and Python classifier and suggests adaptations to the Python classifier, followed by an adapted implementation of the function body.

```
[START OUTLINE]
# Error Analysis
The reported error example shows that the VLM considers `(gripper_around lamp_bulb)` true,
    but the current grounding function returns false. Analyzing the code, the failure likely
     arises from the orientation check, because:
- The gripper orientation roll and pitch are hardcoded as zero.
- The lamp_bulb has roll = -1.75 rad (~ -100deg), pitch = 0.54 rad (~31deg), both far from
    zero.
- Orientation difference tolerance is 0.3 rad (~17deg), so this difference fails the
    orientation check, causing the predicate to return False.

Other checks:
- Gripper closed is False, so pass.
- Distance between gripper and part xy centers looks small enough (gripper at [0.456, 0.183],
     bulb at [0.43, 0.179]) => dx=0.026, dy=0.004 => dist_xy ~0.026, within 0.05 tolerance.
- Vertical difference dz = abs(0.014 - 0.045) = 0.031, < 0.05, passes.

Conclusion: orientation check fails due to assumed gripper orientation zero, but robot
    orientation isn't known, so the check is too strict or invalid with current data.

# Suggested Fixes
- Remove or relax orientation checks because gripper orientation data is unknown / not
    available. This aligns with the comment in code: "If this assumption is too strong, this
     step can be omitted or weakened."
- Keep other checks intact (gripper open, xy distance, vertical height relative to part top).

- Optionally, allow a larger orientation tolerance or skip orientation check completely for
    roll and pitch because gripper orientation is not provided and robot does not restrict
    roll/pitch strictly in grasping parts.
- Possibly rename variable to clarify that gripper orientation is assumed unknown and that
    orientation check is being skipped.
- Keep all distance and gripper open checks mandatory to maintain semantic correctness.

# Fixed Code
```python
```

```
def gripper_around(p: Part, xy_tolerance: float=0.05, z_tolerance: float=0.05) -> bool:
    """
    True if the robot's gripper is positioned around and ready to grasp part p.

    Args:
        p (Part): The part to check against
        xy_tolerance (float): Max allowed horizontal xy distance (m)
        z_tolerance (float): Max allowed vertical distance (m) from gripper to part top
     surface

    Returns:
        bool: True if gripper is horizontally close, vertically near the top of the part,
     and open.
    """
    robot = get_robot()
    # Gripper must be open (ready to grasp)
    if robot.gripper_closed:
        return False

    # Horizontal distance in xy between gripper center and part center
    dx = robot.gripper_center[0] - p.center[0]
    dy = robot.gripper_center[1] - p.center[1]
    dist_xy = math.sqrt(dx * dx + dy * dy)
    if dist_xy > xy_tolerance:
        return False

    # Vertical position check: gripper height close to part top (bounding_box z_max)
    part_top_z = p.bounding_box[5]
    dz = abs(robot.gripper_center[2] - part_top_z)
    if dz > z_tolerance:
        return False

    # Orientation check omitted due to unknown gripper orientation

    return True
```
```
# Grounder Description
This grounding function checks whether the robot gripper is open and located approximately
    horizontally (within 5 cm) above the center of the part, and vertically near (within 5
    cm) the top surface of the part, making it properly positioned "around" the part ready
    for grasping. The function assumes gripper orientation roll and pitch unknown and does
    not enforce orientation alignment.
[END OUTLINE]
```

We only retain the adapted Python classifier if it not yields a worse F1 score $\mathcal{F}$ on the interaction dataset, thereby mitigating hallucinations.

## F   WONG ET AL. - LEARNING DOMAIN MODELS

Wong et al. (2023) propose to learn domain models by generating a large set of operator candidates. This set is then filtered by the ones that are used to solve a predefined set of planning problems. Wong et al. (2023) redo the operator sampling two times, which we also used for evaluating it on the IPC domains. However, for both IPC domains, their method was not able to generate a set of operators that solved the planning for any of the tasks. Therefore, all operators were discarded, resulting in an empty set of operators. We used the official implementation from Wong et al. (2023).

## G   IPC DOMAIN

In the following, we will list natural language instructions and sample plans for some tasks in the *logistics* and *household* domains.

An exemplary skill in the *household* domain with annotations looks like following:

```
def heat_food_with_pan(food: str, pan: str):
    "This action enables the robot to heat food which is heatable with a pan. The food should
     be placed on the pan, and the pan needs to be placed on a stove burner before executing
     this action. Note that the food is no longer pickupable after it has been heated."
```

## G.1 LOGISTICS

Task 1: Transport package `package_0` to location `location_2`

```
load_truck('package_0', 'truck_1')
fly_plane('plane_0', 'location_1', 'location_0')
unload_truck('package_0', 'truck_1')
load_plane('package_0', 'plane_0')
fly_plane('plane_0', 'location_0', 'location_1')
unload_plane('package_0', 'plane_0')
drive_truck('truck_0', 'location_2', 'location_1')
load_truck('package_0', 'truck_0')
drive_truck('truck_0', 'location_1', 'location_2')
unload_truck('package_0', 'truck_0')
```

Task 2: Transport package `package_3` to location `location_3`, package `package_4` to location `location_0`, package `package_1` to location `location_2`, package `package_0` to location `location_0` and package `package_2` to location `location_4`

```
load_truck('package_1', 'truck_1')
load_plane('package_4', 'plane_0')
load_plane('package_0', 'plane_0')
drive_truck('truck_1', 'location_4', 'location_5')
unload_truck('package_1', 'truck_1')
load_plane('package_1', 'plane_0')
fly_plane('plane_0', 'location_5', 'location_2')
unload_plane('package_4', 'plane_0')
load_truck('package_4', 'truck_0')
load_plane('package_3', 'plane_0')
unload_plane('package_1', 'plane_0')
unload_plane('package_0', 'plane_0')
load_truck('package_0', 'truck_0')
drive_truck('truck_0', 'location_2', 'location_0')
unload_truck('package_4', 'truck_0')
unload_truck('package_0', 'truck_0')
drive_truck('truck_0', 'location_0', 'location_1')
load_truck('package_2', 'truck_0')
drive_truck('truck_0', 'location_1', 'location_2')
unload_truck('package_2', 'truck_0')
load_plane('package_2', 'plane_0')
fly_plane('plane_0', 'location_2', 'location_5')
unload_plane('package_3', 'plane_0')
load_truck('package_3', 'truck_1')
unload_plane('package_2', 'plane_0')
load_truck('package_2', 'truck_1')
drive_truck('truck_1', 'location_5', 'location_3')
unload_truck('package_3', 'truck_1')
drive_truck('truck_1', 'location_3', 'location_4')
unload_truck('package_2', 'truck_1')
```

## G.2 HOUSEHOLD

Task 1: put `apple_2` on `side_table_2`

```
go_to("dining_table_1")
put_on("mug_1", "dining_table_1")
go_to("fridge_1")
open("fridge_1")
pick_up("lunch_box_2", "fridge_1")
go_to("dining_table_1")
put_on("lunch_box_2", "dining_table_1")
open_small("lunch_box_2")
pick_up_from("apple_2", "lunch_box_2")
go_to("side_table_2")
put_on("apple_2", "side_table_2")
```

Task 2: heat `pizza_1` with `pan_2`, and put it on `plate_2`

```
go_to("dining_table_1")
put_on("mug_1", "dining_table_1")
go_to("cabinet_2")
open("cabinet_2")
pick_up("pizza_box_1", "cabinet_2")
go_to("countertop_2")
put_on("pizza_box_1", "countertop_2")
open_small("pizza_box_1")
```

```
go_to("drawer_1")
pick_up("pan_2", "drawer_1")
go_to("countertop_2")
put_on("pan_2", "countertop_2")
transfer("pizza_1", "pizza_box_1", "pan_2")
pick_up("pan_2", "countertop_2")
go_to("stove_burner_2")
put_on("pan_2", "stove_burner_2")
heat("pizza_1", "pan_2")
go_to("cabinet_3")
open("cabinet_3")
pick_up("plate_2", "cabinet_3")
go_to("countertop_2")
put_on("plate_2", "countertop_2")
go_to("stove_burner_2")
pick_up("pan_2", "stove_burner_2")
go_to("countertop_2")
put_on("pan_2", "countertop_2")
transfer("pizza_1", "pan_2", "plate_2")
```

## H   GENERALIZED EXPLORATION WALK

We evaluate the domain models by computing an adapted version of the EW metric of Mahdavi et al. (2024). It contains two adaptations:

**(1) Evaluating context-centric domain models with the EW scores.**   Since LODGE learns not a full domain model, but context-centric domain models, the final domain model does not contain operators for all aspects of the domain. For example for setting a table, LODGE would not learn operators for wiping the floor. Consequently, we have to adapt the EW score do evaluate context-centric domain models. For every task, we bootstrap the generated and predefined domain model to only contain operators required to solve that task. We evaluate the EW score on this task-centric domain model. We do this for all tasks and average the metrics.

**(2) Sampling over operators instead of actions**   When sampling a random walk from a domain model, Mahdavi et al. (2024) sample the next action uniformly from all available actions. With this, operators that can be executed with many different objects dominate the EW score. For example for the *household* domain, the agent can always *go-to* another furniture. The sampled random walk consequently contains mostly *go-to* actions and only rarely evaluates the similarity of other operators. To fix this, we sample uniformly over the applicable operators and then sample uniformly over all actions from the selected operator. With that, all operators are evaluated evenly.

Additionally, we sample 500 plans compared to 100 used by Mahdavi et al. (2024).

## I   LODGE PROMPTS

### I.1   LLM_DOMAIN

System

```
You are a planning expert tasked with developing a world model for planning based on a user
    instruction.

---

<!-- when generating content for the sections listed below, follow the specified format
    exactly. You can leaf out sections you don't need. -->
### Explanation
<!-- task specific explanation and chain-of-thought reasoning -->

### Change/Add Action(s)
1. {action-name-1}: {(add|edit|delete)}
    - Description: {description what happens during the action}
    - PDDL Definition:
        ```pddl
        (:action {pddl_action_name}
            :parameters {pddl_action_parameters}
            :precondition {pddl_action_preconds}
            :effect {pddl_action_effects}
```

```
        )
        ```

### Change/Add Predicate Definitions
<!-- 'state' predicates describe the current situation of objects in the world. Their
      description must specify the conditions under which they hold, so a classifier can later
      decide from perception data (e.g., 6D poses) whether the predicate is true.-->
- ({predicate_name} {predicate_args...}): {state|other}. {predicate_description}

### Change Initial State
<!-- add predicates that should be changed, without text, leave 'None' if no change -->
{predicate_4}: {true|false|remove}
{predicate_8}: {true|false|remove}

### Change Goal State
<!-- add predicates that should be changed, without text, leave 'None' if no change -->
{predicate_7}: {true|false|remove}
```

## User

```
### User Instruction
assemble the lamp

### Domain Knowledge
{domain description}

### Predicates
- (assembled ?obj1 - part ?obj2 - part). other: Evaluates whether obj1 has been assembled
      with obj2 the way they should be, e.g. when obj1 has been screwed into obj2, or obj1 has
      been put on top of obj2.

### Actions

### Types
robot table part - object

### Objects
arm - robot table - table lamp_base lamp_bulb lamp_hood - part

### Skill Library
- def hover_above_part(part: str):
- def set_gripper_around_part(part: str):
- def move_linear_up():
- def open_gripper():
- def close_gripper():
- def align_orientation_for_assembly(held_part: str, fixed_part: str):
- def screw_touching_parts_together(held_part: str, fixed_part: str):
- def move_linear_down_until_touching(held_part: str, fixed_part: str):

### Initial State
init state will be replaced

### Your Task
1. Define the goal: Based on the user instruction, create a PDDL goal that reflects the
      objective the user wants to accomplish.
2. List of predicates: Inspect the predicates that will be used to describe the state of the
      world and the relationships between entities in the domain.
3. Define actions:
    - Based on the goal and available skills in the skill library, define a set of PDDL
      actions that enable planning toward the goal.
    - Preferably define high-level actions that abstract over one or more low-level skills
      to support hierarchical planning, but don't add object specific actions, e.g. 'unstack-
      block-2'.
    - Each PDDL action should:
        - Have a clear and descriptive name
        - Include a general-purpose description (not instance-specific)
        - Include a PDDL definition: (:action <action_name> :parameters <parameters> :
      precondition <precondition> :effect <effect>)
    - You are encouraged to:
        - Define high-level composite actions when they simplify planning.
        - Define individual PDDL actions for each skill you intend to use that has no PDDL
      action yet.
        - If PDDL actions are already given in the user prompt, use them and only add new
      ones if necessary.
        - Avoid unnecessary actions - only include those essential to achieving the goal.
        - Don't create object specific actions, e.g. 'switch-block-2-and-block-3', but rather
      generate a general action 'switch-blocks'.
        - Temporal ordering of subtasks can later be enforced by the LLM-planner
```

Assistant

```
### Explanation
The goal is to assemble the lamp; (...)

### Change/Add Action(s)
1. grasp_part: add
    - Description: Robot arm hovers above a part, grasps it by closing the gripper, and
     lifts it up.
    - PDDL Definition:
    ```pddl
    (:action grasp_part
        :parameters (?r - robot ?p - part ?t - table)
        :precondition (and (not (assembled ?p ?_)) ; part is free to grasp (not already
     assembled, simplified)
                        )
        :effect (and (holding ?r ?p)
                        (not (on_table ?p ?t))
                        )
    )
    ```

(...)

4. assemble_parts: add
    - Description: Assemble two parts by screwing them together while the robot holds one
     part aligned and touching the other.
    - PDDL Definition:
    ```pddl
    (:action assemble_parts
        :parameters (?r - robot ?part1 - part ?part2 - part)
        :precondition (and (holding ?r ?part1)
                        (touching ?part1 ?part2)
                        )
        :effect (and (assembled ?part1 ?part2)
                    (not (holding ?r ?part1))
                    )
    )
    ```

### Change/Add Predicate Definitions
- (holding ?r - robot ?p - part): state. The robot is currently holding the part with its
    gripper.
- (on_table ?p - part ?t - table): state. The part is resting on the table.
- (aligned ?p1 - part ?p2 - part): state. The held part is aligned with the fixed part for
    assembly (xy position & roll/pitch orientation aligned).
- (touching ?p1 - part ?p2 - part): state. The held part is in contact with the fixed part,
    ready for assembly.

### Change Goal State
(assembled lamp_bulb lamp_base): true
(assembled lamp_hood lamp_base): true
```

## I.2  LLMTRANSLATE

User

```
### Predefined Skills
```python
def hover_above_part(part: str): ...
def set_gripper_around_part(part: str): ...
def move_linear_up(): ...
def open_gripper(): ...
def close_gripper(): ...
def align_orientation_for_assembly(held_part: str, fixed_part: str): ...
def screw_touching_parts_together(held_part: str, fixed_part: str): ...
def move_linear_down_until_touching(held_part: str, fixed_part: str): ...
```

### Objects
arm - robot table - table lamp_base lamp_bulb lamp_hood - part

### Predicate Definitions
(assembled ?obj1 - part ?obj2 - part): (...)
(...)
(touching ?p1 - part ?p2 - part): (...)

### PDDL Action Definition
```

```
grasp-part: add
    - Description: Robot arm hovers above a part, grasps it by closing the gripper, and
     lifts it up.
    - PDDL Definition:
        ```pddl
        (:action grasp-part
            :parameters (?r - robot ?p - part ?t - table)
            :precondition (and (on_table ?p ?t) (not (holding ?r ?p)))
            :effect (and (holding ?r ?p) (not (on_table ?p ?t)))
        )
        ```

### Task
The current state fulfills the PDDL action 'grasp-part''s preconditions. Your task is to
    propose a sequence of predefined skills to reach the effect of the PDDL action.

Here is an outline of what your response should look like:
[START OUTLINE]
# Outline Current State
[given the action preconditions, outline the current state]
# Action Description
[insert your analysis what the PDDL action is trying to achieve based on the current state
    and the effects that must be reached]
# Skill Mapping
[insert a bullet list of predefined skills as '<function_name>(<arg1>, <arg2>, ...)' that,
    starting from the current state, reach the effects. Don't include skills to 'confirm'
    the preconditions. Use variables 'from variables import r_robot, p_part, t_table' to
    reference pddl parameters, or strings if you need additional arguments where no pddl
    parameter exists]
[END OUTLINE]
```

### Assistant

```
[START OUTLINE]
# Outline Current State
- The robot arm (r_robot) is not holding the part (p_part).
- The part (p_part) is on the table (t_table).

# Action Description
- The grasp-part action aims for the robot arm to pick up the part from the table.
- The preconditions require the part to be on the table and not already held.
- The effect is that the robot will be holding the part and it will no longer be on the
    table.
- To achieve this, the robot needs to position its gripper over the part, close the gripper
    around the part, and then lift the part up from the table.

# Skill Mapping
- hover_above_part(p_part)
- set_gripper_around_part(p_part)
- close_gripper()
- move_linear_up()
[END OUTLINE]
```

## I.3 LLM_DECOMPOSE

### User

```
You are given a high-level PDDL action and supporting context. Your task is to decompose
    this high-level action into a set of meaningful, lower-level PDDL actions that result in
     the same effect.

### High-level Action '(grasp-part arm lamp_bulb table)'
grasp-part: add
    - Description: Robot arm hovers above a part, grasps it by closing the gripper, and
     lifts it up.
    - PDDL Definition:
        ```pddl
        (:action grasp-part
            :parameters (?r - robot ?p - part ?t - table)
            :precondition (and (on_table ?p ?t) (not (holding ?r ?p)))
            :effect (and (holding ?r ?p) (not (on_table ?p ?t)))
        )
        ```

### Predicates
{predicates}

### Actions
```

```
### Types
robot table part - object

### Objects
arm - robot table - table lamp_base lamp_bulb lamp_hood - part

### Initial State
(on_table lamp_hood table) (on_table lamp_base table) (on_table lamp_bulb table)

### Goal State
(and (not (on_table lamp_bulb table)) (holding arm lamp_bulb))

### Skill Library
- def hover_above_part(part: str):
(...)

### Suggested Decomposition
```pddl
hover_above_part(p_part)
set_gripper_around_part(p_part)
close_gripper()
move_linear_up()
```

---

### Instructions
Follow the steps below to complete the decomposition:
1. Describe the Initial State
2. Understand the High-Level Action: Examine the :precondition and :effect. Identify what
   state change it induces.
3. Define Lower-Level Actions (add new predicates if necessary): Construct new actions that
   together implement the high-level action.
   - Refer to the suggested decomposition for a LLM-proposed skill sequence. Your actions
   should cover the skill sequence.
   - Only include actions that result in a state change and are not already defined below #
    Actions.
   - Do not include meta-actions like confirm, complete, sense, or computation-related
   steps.
   - Construct new predicates that are required to capture the necessary conditions and
   effects of the lower-level actions.
   - Define individual PDDL actions for each skill you intend to use.
4. Specify the Goal State: List the predicates that reflect the intended outcome of the high-
   level action.

{syntax}
```

## Assistant

```
### Explanation

The high-level action '(grasp-part arm lamp_bulb table)' changes the state from the lamp
    bulb being on the table to the robot arm holding it. The precondition requires the part
     to be on the table and not currently held; the effect requires that the robot is holding
     the part and that the part is no longer on the table.

To decompose this into lower-level PDDL actions, the suggested skill sequence is:

1. hover above the part
2. set the gripper around the part
3. close the gripper (to grasp)
4. lift the part by moving the arm linearly up

### Change/Add Predicate Definitions

- (hovering_above ?r - robot ?p - part): state. The robot ?r is currently positioned
    hovering above part ?p.
- (gripper_around ?r - robot ?p - part): state. The robot's gripper is opened and correctly
    positioned around part ?p.
- (gripper_closed ?r - robot): state. The robot's gripper is closed.
- (holding ?r - robot ?p - part): state. The robot is holding the part with its gripper. (
    already given)
- (on_table ?p - part ?t - table): state. The part rests on the table. (already given)

### Change/Add Action(s)

1. hover-above-part: add
     - Description: The robot moves and hovers above the specified part.
```

```pddl
        - PDDL Definition:
            ```pddl
            (:action hover-above-part
                :parameters (?r - robot ?p - part ?t - table)
                :precondition (on_table ?p ?t)
                :effect (hovering_above ?r ?p)
            )
            ```

2. set-gripper-around-part: add
    - Description: The robot opens its gripper and positions it around the part to grasp.
     Requires hovering over the part.
    - PDDL Definition:
        ```pddl
        (:action set-gripper-around-part
            :parameters (?r - robot ?p - part)
            :precondition (hovering_above ?r ?p)
            :effect (and (gripper_around ?r ?p) (not (gripper_closed ?r)))
        )
        ```

3. close-gripper: add
    - Description: The robot closes the gripper, grasping the part that is around its gripper
    . Requires the gripper to be around the part.
    - PDDL Definition:
        ```pddl
        (:action close-gripper
            :parameters (?r - robot ?p - part)
            :precondition (gripper_around ?r ?p)
            :effect (and (holding ?r ?p) (gripper_closed ?r) (not (gripper_around ?r ?p)))
        )
        ```

4. move-linear-up: add
    - Description: The robot lifts the held part by moving linearly up, causing the part to
     no longer be on the table. Requires holding the part.
    - PDDL Definition:
        ```pddl
        (:action move-linear-up
            :parameters (?r - robot ?p - part ?t - table)
            :precondition (holding ?r ?p)
            :effect (not (on_table ?p ?t))
        )
        ```
```

## I.4   LLM~REASONER~

```
You are given a decomposition hierarchy and a record of skills executed in a simulated
    environment. The last skill has failed during execution. Your goal is to identify why
    the observed effect of the simulation diverged from the expected effect of that skill.
    The simulation and skill implementations are correct and fixed - you cannot modify them.
     Your focus is on reasoning about the planning model and its action decomposition.

---

### Context:
{domain description}

State before (set-gripper-around-part-lowlevel arm lamp_bulb):
(on_table lamp_base table)
(on_table lamp_bulb table)
(on_table lamp_hood table)
(hovering_above arm lamp_bulb)

Decomposition Hierarchy
- (grasp-part arm lamp_bulb table)
    - (hover-above-part arm lamp_bulb table): hover_above_part('lamp_bulb')
    - (set-gripper-around-part arm lamp_bulb)
       - (set-gripper-around-part-lowlevel arm lamp_bulb)

Current Operator
(:action set-gripper-around-part-lowlevel
    :parameters (?r - robot ?p - part)
    :precondition (and (hovering_above ?r ?p) (not (gripper_closed ?r)))
    :effect (gripper_around ?r ?p)
)

Executed Python Skill for PDDL Action `(set-gripper-around-part-lowlevel arm lamp_bulb)`:
```

```
1512
1513     set_gripper_around_part('lamp_bulb')
1514
         Execution Failure:
1515     Expected Change:
         - (gripper_around arm lamp_bulb): False -> True
1516
1517     Ground Truth Change:
         - (hovering_above arm lamp_bulb): True -> False
1518     - (gripper_around arm lamp_bulb): False -> True
1519
         ---
1520
1521     Your Task:
         1. Summarize the state before executing the failed action.
1522     1. How did the environment change by executing set_gripper_around_part('lamp_bulb').
         2. Identify the cause of the deviation
1523        - Looking at the decomposition hierarchy, are skills missing before executing
1524          set_gripper_around_part('lamp_bulb') to ensure a successful execution (i.e. no collision
              or undefined behavior)?
1525             - If so:
1526                 - which skills should have been executed before set_gripper_around_part('
             lamp_bulb') and can the preconditions be tightened to ensure that?
1527            - else:
1528                - should actions be removed?
                   - why are the expected effects different and how must the PDDL action set-gripper
1529         -around-part-lowlevel be changed to realign it with the ground truth change?: Adapt the
             PDDL action.
1530               - should additional predicates be invented to capture the effects more accurately
1531         ?
1532               - are preconditions missing in any of the PDDL action definitions?
                   - or, were any skills redundant, missing, or incorrectly ordered?
1533         Note that:
1534         - We use FastDownward Planning, so naming does not impact the plan, only forming correct
              PDDL definitions does.
1535         - Do not assume the current expected effects are correct - critically assess and revise
1536          them as needed
             - Adapt the PDDL actions in a general way, e.g. by using conditional effects where
1537          applicable ('when', 'forall', 'imply', ...)
1538
```

## Assistant response

```
1540
1541     1. **Summary of State Before Executing the Failed Action**
1542     - The robot arm ('arm') is *hovering above* the 'lamp_bulb'.
1543     - The gripper is *not closed*.
         - The gripper is *not around* the 'lamp_bulb'.
1544     - None of the parts are held.
         - All parts ('lamp_base', 'lamp_bulb', 'lamp_hood') are on the table.
1545     - No parts are aligned, touching, or assembled.
         - The robot is *not* hovering above any part except the 'lamp_bulb'.
1546     - The gripper is *not around* any part at this point.
1547
1548     2. **How Did the Environment Change by Executing 'set_gripper_around_part('lamp_bulb')'**
1549     - The predicate '(gripper_around arm lamp_bulb)' changed from 'False' to 'True', as expected.
1550
         - Unexpectedly, the predicate '(hovering_above arm lamp_bulb)' changed from 'True' to 'False
1551      '.
1552
         This means that while the gripper correctly adjusted to be around the lamp bulb, the robot
1553         stopped hovering above it as a side effect.
1554
1555     3. **Identification of the Cause of the Deviation**
1556     - The PDDL action for 'set-gripper-around-part-lowlevel' is:
1557         ```pddl
1558         (:action set-gripper-around-part-lowlevel
             :parameters (?r - robot ?p - part)
1559             :precondition (and (hovering_above ?r ?p) (not (gripper_closed ?r)))
1560             :effect (gripper_around ?r ?p)
         )
1561         ```
1562
         (...)
1563
1564     ### **Summary and Recommendation**
1565     - The main reason for the deviation is an incomplete PDDL effect specification in 'set-
             gripper-around-part-lowlevel'.
```

```
- The robot physically must move from a hovering pose to a gripper-encompassing pose, which
    invalidates 'hovering_above' for the part.
- Add '(not (hovering_above ?r ?p))' to the effects of the action to align the expected
    effects with actual behavior.
- No additional skills or predicates are needed, but the planner must be aware that hovering
    and gripper placement are mutually exclusive states.
```

## User prompt to get decision of reasoner

```
Determine the most probable fix type based on the following options:
- one of the action definitions listed in 'decomposition hierarchy' must be corrected: 'pddl-
    fix'
- some skills should be executed before the action: 'prior-skills': list the
- the skill was incorrectly instantiated or used: 'incorrect-instantiation'
- alternatively, the pddl action (set-gripper-around-part-lowlevel arm lamp_bulb) should be
    implemented with multiple skills: 'multiple-skills'

Independent of the chosen fix type, list ALL operators that must be edited to resolve the
    issue. If multiple operators must be changed, list them comma-separated ([op1, op2,
    ...]).

Output Format:
```json
{
    "type_of_fix": "<chosen-fix-type>",
    "operators": ["<corrected-action>", "..."]
}
```
```

## Final decision of reasoner

```
```json
{
    "type_of_fix": "pddl-fix",
    "operators": ["set-gripper-around-part-lowlevel"]
}
```
```

