# OpenReview forum: "Learning Hierarchical Domain Models through Environment-Grounded Interaction"
_ICLR.cc/2026/Conference — Submitted to ICLR 2026_

### Official Review · Reviewer_p9vb · 2025-10-29

**Soundness:** 3
**Presentation:** 2
**Contribution:** 3
**Rating:** 4
**Confidence:** 2

**Summary:**

This paper introduces LODGE, a framework for learning hierarchical planning domain models using LLMs and environment-grounded interaction. LODGE autonomously generates, refines, and empirically grounds domain knowledge through hierarchical abstraction, operator decomposition, predicate invention, and data-driven learning of predicate classifiers. Its core novelty lies in avoiding reliance on human feedback or prior skill annotation. The framework is evaluated on two International Planning Competition domains and a robotic assembly benchmark.

**Strengths:**

1. This paper investigates an important problem: how to enable agents to automatically generalize to new domains without human annotations.

2. The idea is sound, and the experimental results demonstrate its effectiveness.

**Weaknesses:**

1. Scalability: Can LODGE be extended to more complex tasks or dynamically changing environments? How does it handle partially observed environments? Moreover, the experiments in this paper are conducted in simulated environments, which allow LODGE to perform frequent trial and error. How can LODGE be adapted to the real world, where trial-and-error costs are much higher and the environment is difficult to reset?

2. Dependence on model capability: From the experimental results, it can be seen that although a self-correction mechanism is proposed, the LLM may still generate incorrect operators, predicates, or code, particularly when using smaller models (e.g., GPT-4o mini), which struggle with handling "side effects". In addition, the classifier's learning relies on pseudo-labels generated by the VLM. If the VLM labels are inaccurate, they could mislead the classifier’s optimization.

3. Although LODGE does not rely on manual annotations, the feedback process typically incurs a large token cost. Could the authors compare the total token consumption of LODGE with that of the baselines (e.g., Mahdavi)?

If the authors can address these points, I would raise my score.

**Questions:**

1. Can LODGE be combined with manual annotations, for example, using a small amount of human-labeled data to improve its learning efficiency?

---

> ### Author Response · Authors · 2025-11-21
>
> We thank you for the review and want to answer in the questions raised about scalability and model capability.
>
> **Scalability**
> We specifically think that one benefit of LODGE is to better scale to complex tasks than related methods. For once, LODGE is robust to an increase in the skill set size due to the task-centric domain learning we propose, which focuses on learning the domain model for the relevant skills required to solve a given task. Apart from that, the hierarchical decomposition of the domain model scales better to larger domains, by distributing the operators to be learned across multiple hierarchy levels and sub-domains. Especially when integrating LLMs, this is useful, since the LLM has to attend to fewer operators at the same time. With the FurnitureBench and Household domains, we deliberatively chose complex domains to evaluate the ability of LODGE to learn suitable domain models. We believe that these domains are good representatives for complex problems, due to the spatial and logical reasoning required.
>
> *Dynamic Environments:*
> LODGE can indeed be extended for dynamic environments. The environment grounding with predicate classifiers in LODGE naturally detects misalignments between the believed and observed effects. While we solely use the misalignment to refine the operators, an extension could directly trigger replanning or a more sophisticated method that tries to distinguish modelling errors from failures. Replanning can be implemented by using the observed effect as new `start state` and restarting planning to the goal from that state.
>
> *Partial Observability:*
> We do believe that learning in partially observable environments is an important property. However, as most work in this domain, we currently do assume a fully observable environment to limit the scope of the work. However, extensions of LODGE to partial observability are very interesting directions to extend LODGE in future work.
>
> *Human Annotations:*
> Motivated by the open-world setting, we specifically focused LODGE on learning domain models in a sample efficient manner, by limiting the *plan rollouts* to 10.
> Therefore, learning the domain model solely in the real world would be possible, with slight adaptions. While it is possible to rely on a human to reset the environment during domain learning after *plan rollout*, it would also be possible to not reset the environment, but recover learning from the new state as new start state (assuming that the environment is recoverable by the robot).
> What mostly prevents LODGE currently from learning in the real world is the possibility of an invalid plan during rollout causing a collision, which could be prevented by a user double-checking the steps before execution, or by the inclusion of a collision-free motion planner as part of the robot control stack.
>
> **Model Capability**
> We are aware that LODGE does not provide theoretical guarantees about the accuracy of the learned domain model, when specifically weaker models generate invalid operators or code.
> We do not use the most powerful models, but evaluate on models that can be self-hosted by academic institutions. Also, we see scaling effects from older to newer models in the evaluation, which promises that future models may yield more robust generation by default.
> Additionally, we observed frequent VLM hallucinations when classifying the state. This motivated us to not directly use them for classifying the state, but only as a *guidance* to check and refine the code-based classifier with the methods we propose, while considering imperfect classifications. Firstly, for the hyperparameter optimization, we define a *trust region* around the default hyperparameters, which avoids overfitting to *unplausible* hyperparameters, especially with small data at the beginning of learning.
> Secondly, we re-prompt the LLM to update the classifier code only for severe misalignments between the VLM classification and the code-based classification. Lastly, we only retain the self-refinement or hyperparameter optimization if it improves the evaluation score on the pseudo-annotated dataset.
>
> **Human Labeled Data**
> Human labeled data can indeed speed up domain learning: Passing additional information to the LLM that generates operators improves the initial correctness of the operators. While LODGE learns the domain model online during planning and execution of plans in the environment, it could be adapted to learn the domain model offline first from small human-annotated trajectories, e.g. by traversing the trajectories and verifying the alignment of the domain model with the annotated states. While we focus on a automated system that does not rely on human annotations, analyzing its impact on the learning is an interesting future direction.
>
> **Token Usage**
> We see that comparing the LLM resources for every task is an important ablation. We added a comparison in the ablations of the revised paper (section A.1 of the appendix).

---

### Official Review · Reviewer_MNNh · 2025-10-30

**Soundness:** 2
**Presentation:** 2
**Contribution:** 2
**Rating:** 2
**Confidence:** 3

**Summary:**

The paper proposes an autonomous domain learning framework that generates hierarchical PDDL domains and corrects inconsistencies using environment feedback. The proposed method is evaluated on two IPC domains and a robotic assembly domain. While the overall idea is intriguing, the novelties are weak in general. Critical experiments are missing as detailed in the following weakness session.

**Strengths:**

1. The paper is well written overall. The messages are conveyed clearly.
2. The proposed hierarchical domain models can be interesting if supported by more thorough experiments.

**Weaknesses:**

The three novelties claimed by this paper are fairly weak for the following reasons:

1. Novelty 1 "Domain learning from planning feedback": While this paper claims only minimum prior knowledge is required, the initial domain and some critical predicates are provided by the user as shown in Appendix H1. The idea of using model-based environment feedback to optimize predicates is similar to the prior paper [1], which compromises the novelty of this paper.

2. Novelty 2 "Hierarchical domain models": This paper proposes a hierarchical domain generation framework that iteratively breakdown the high-level operator (e.g., grasp(bulb)) to lower-level operators like approach(bulb) etc. However, using multi-level operators with finer abstraction will significantly increase the complexity of PDDL searching, which might degrade the success rate of task planning. Trade-off between abstraction levels and planning success is not discussed. There is no experiment result to support that the proposed hierarchical PDDL domain promote planning success rate.

3. Novel 3 "Predicate invention and online classifier learning": The idea is very close to the paper [1]. Although this paper uses slightly-different pseudo-labeled environment interaction, it is not compared to the model-based feedback used in [1]. Critical ablation study is missing.

4. Real robot experiment is missing. Given the noise in object localization and less accurate feedback from the proposed pseudo-labeled environment interactions, the proposed method will face serious challenges in real robot experiment.

5. Typos and grammar errors:
Line 253: ".If"
Line 283, 284: "f1 score" should be "F1 score"

[1] VisualPredicator: Learning Abstract World Models with Neuro-Symbolic Predicates for Robot Planning https://arxiv.org/abs/2410.23156

**Questions:**

1. Can you provide experiment results of planning success rate using finer operators from the proposed hierarchical domain versus high-level operators?
2. Only predicate grounding results are provided in Table 2. Can you provide planning success rates of the proposed method versus InterPreT and Pix2Pred?
3. Have you tried to run any real robot experiment?

---

> ### Author Response · Authors · 2025-11-21
>
> We thank you for the feedback.
> We want to clarify the novelty of our three contributions in the following.
>
> **Domain Learning from Planning Feedback**
> We do indeed assume minimal prior knowledge of the domain model we want to learn. Specifically, we only assume `goal predicates` and `object types` given, while the remaining part of the domain, namely predicates and all operators, are learned.
> Assuming `goal predicates` (e.g. `assembled` in H1) is common practice (also assumed in [1]), as a way of specifying what constitutes a successful task execution. Object types are in general assumed too in recent methods, e.g. [1] defines them in the definition of the environment. The prompt in H1 only injects these two things, whereby the LLM constructs the predicates and operators. Moreover, in contrast to recent method for domain learning, we do not assume skill annotations (Guan et al.) or operator signatures to be given (Mahdavi et al., to compute exploration walk metric).
>
> We would appreciate it if the reviewer could clarify whether this aligns with their understanding of “initial domain and some critical predicates provided by the user,” so that we can address any remaining concerns accurately.
>
> We have to hold against the claim that our method of interacting with the environment is similar to the model-based environment feedback in [1] to optimize the predicates. [1] use the environment interaction to determine a robust set of predicates, grounded by a VLM. LODGE on the other hand primarily hinges on environment feedback to verify and refine operators. We do use a VLM to ground predicates, which is inspired by [1]. However, our method uses these groundings to refine scripted classifiers, while [1] selects  the *most-reliable* set of predicates with the sample-and-select algorithm [1]. Our approach on predicate grounding avoids relying on a VLM in-the-loop for classification.
> Apart from this, the environment interaction is mostly used to verify and refine operators from environment feedback, which is one main contribution of the work.
>
> **Hierarchical Domain Models**
>
> We appreciate the reviewer’s concern regarding the potential increase in planning complexity due to hierarchical domain models. Our formulation of hierarchical domain models is indeed novel. While hierarchical modeling can introduce additional considerations, in our approach it does not increase planning complexity because the planner treats each operator, whether high-level or primitive, uniformly. Specifically, abstract operators are represented in a way that allows planning on a single hierarchy level without requiring decomposition of high-level operators (due to the downward refinement property [B]). Apart from that, our contibution lies in automatically learning *when to decompose*, where the concrete decomposition is planned and not predefined in the domain model.
>
> We argue that decomposition is effective in splitting large domain models into manageable units, improving the scalability of domain learning. We stem on previous works in the hierarchical planning community that demonstrate the computation benefits of using hierarchical abstractions [C, D] to reduce, or distribute, planning complexity across multiple, simpler planning steps.
>
> **Predicate Invention**
> The predicate invention proposed in [1] is very close to [A], whereby [A] focuses on online learning, while [1] learns predicates offline.
> The VLM-based predicate grounding in LODGE is indeed inspired by [A]. However, our contribution lies in the algorithm that uses these pseudo-labels to learn a more accurate, coded classifier. We show that this yields more accurate classifiers that also do not depend on VLM groundings for classification.
>
> Both [1] and [A] use a version of the cluster-and-intersect algorithm to construct operators from grounded predicates. Since the code for [1] and [A] is not publicly available, direct comparison is difficult. Instead, we compared our method to the originally introduced cluster-and-intersect algorithm in Table 4. Essentially, `C&S w. VLM` is very similar to [A], but without predicate subselection or their adaptation of the cluster-and-intersect algorithm.
> `C&S w. Py.` applies the original cluster-and-intersect algorithm while assuming a perfect grounder without misclassifications. As [A] introduces predicate subselection and an adapted algorithm specifically to improve robustness to VLM misclassifications, its domain-level performance would be expected to lie between `C&S w. VLM` and `C&S w. Py`.
>
> [A] A. Athalye et al., “From Pixels to Predicates: Learning Symbolic World Models via Pretrained Vision-Language Models,” arXiv.2501.00296.
> [B] F. Bacchus and Q. Yang, “The Downward Refinement Property.,”. Available: https://www.ijcai.org/Proceedings/91-1/Papers/045.pdf
> [C] Simon, H. 1962. The architecture of complexity. Proc. American Philosophical Society 106
> [D] Yang, Q. 1990. Formalizing planning knowledge for hierarchical planning. Comput. Intell. 6

---

> > ### Comment · Reviewer_MNNh · 2025-11-27
> >
> > The author's reply addressed my concerns in prior knowledge requirement and planning complexity, but the missing experiments in my questions are not touched. I will raise my rating accordingly.

---

> ### Author Response · Authors · 2025-12-03
>
> Thank you for your response and for highlighting the importance of real world experiments. In response, we added two more experiments on a real Franka robot for the FurnitureBench lamp assembly. Both are now included in the manuscript (main text and appendix).

---

### Official Review · Reviewer_3u2N · 2025-10-31

**Soundness:** 2
**Presentation:** 3
**Contribution:** 3
**Rating:** 6
**Confidence:** 4

**Summary:**

In this paper, the authors propose LODGE, a framework that uses LLMs to generate domain models in a hierarchical way for planning.  It iteratively decomposes high-level task descriptions into low-level robot skills, invents and refines predicates via Python-based classifiers, and repairs model errors through execution validation. Experiments on 3 domains show that LODGE achieves higher task success rates than baselines.

**Strengths:**

1. This paper aims to automatically generate planning domains to reduce manual engineering efforts, which is a valuable goal for the field. The proposed framework, which contains hierarchical generation and error recovery, is a reasonable design that can effectively improve the accuracy of the generated domain models.
2. The paper is well organized.

**Weaknesses:**

1. The experimental evaluation is limited. Although the FurnitureBench dataset contains multiple assembly tasks, the experiments only report results for the lamp assembly case. To demonstrate the robustness and generalizability of the proposed approach, experiments should be conducted across all available task categories in FurnitureBench. In addition, more tasks, such as those introduced in [1], should be used to further evaluate the method’s general applicability.
2. The paper overlooks several closely related studies that explore the integration of large language models (LLMs) with task planning. For example, [1] proposes leveraging LLMs to infer object affordances, dynamically generate affordance-based domain and problem files, and solve them via a classical planner. [2, 3] discuss how LLMs can refine domain models for open-world planning.
3. The paper should report quantitative metrics on the number of LLM invocations required per task. Such statistics are essential for understanding the efficiency of the proposed approach.

Reference:
[1] AutoGPT+P: Affordance-based Task Planning using Large Language Models, RSS 2024.
[2] Language-augmented Symbolic Planner for Open-world Task Planning, RSS 2024.
[3] Integrating Action Knowledge and LLMs for Task Planning and Situation Handling in Open Worlds, Autonomous Robots 2023.

**Questions:**

1. If an execution failure arises from low-level control errors (rather than incorrect domain models), would the proposed recovery module still be triggered? If so, does this risk incorrectly modify a valid domain model?
2. If the system is allowed to interact with the environment and re-plan without any limits, is it possible to generate a fully accurate domain model? In that case, how many LLM invocations would be required on average?

---

> ### Author Response · Authors · 2025-11-21
>
> We thank you for the feedback and the additional related work. We included them in the related work section of the recently uploaded rebuttal-version of the paper.
> In the following, we want to answer the question about LLM invocations and additional evaluations.
>
> **Number of LLM Invocations per Task**
> We added a new ablation in the appendix of the paper that compares LLM usage of all methods.
>
> **FurnitureBench**
> We agree with the reviewer that evaluating LODGE on the remaining FurnitureBench assembly tasks helps to analyze the generalization of our approach.
> In response, we added following evaluations:
> - *Evaluation on Round Table*: We adapt the domain model learned during the lamp task for the assembly of the round table.
> - *Evaluation on Square Table, Desk, Stool*: We additionally learn the domain model to assemble the square table, desk, and stool. These differ from the lamp by requiring to assemble multiple parts with one part, e.g. four legs with the table top. We learn the domain model on the square table task, and evaluate it on all three tasks.
>
> For the remaing FurnitureBench tasks Drawer, Cabinet and Chair, additional skills (pushing/tight-fit insertion) would be required.
> We do not have these skills available currently, because of which we were unable to evaluate these.
>
> **Questions**
> - Q1: *Trigger low-level control errors the domain refinement?*
>
>     When low-level control errors arise, then yes, we would trigger the recovery module, which currently adapts the domain model based on the misalignment observed. However, the recovery module could be extended to give it the decision whether to refine the domain model, or to trigger re-planning (like a no-op), to avoid incorrectly modifying the domain model. A replanning is naturally possible by discarding the current plan candidate, setting the current state as new start state and replanning from there.
>
> - Q2: *Will we learn a fully accurate domain model with infinite retries?*
>
>     While increasing the environment interactions would increase the probability of generating fully accurate domain models some time, there is no theoretical guarantee for it. The LLM we use for generating the domain and refining it in LODGE might sometimes *not be able* to fix the domain model. Our work focuses on learning accurate domain models with few interactions and little domain knowledge available. This could be combined with a learning based method to further refine the model once we reach a plateau in learning the domain model with LODGE.

---

### Official Review · Reviewer_PUwn · 2025-11-01

**Soundness:** 2
**Presentation:** 2
**Contribution:** 2
**Rating:** 2
**Confidence:** 3

**Summary:**

The paper proposes LODGE, a framework designed to refine planning models generated using LLMs. The paper particularly focuses on robotics settings, where the system has access to the low-level state and a set of primitive actions. Here, the grounding of the predicates generated by the system is performed through classifiers coded up as Python functions, which are refined via a dataset generated through pseudo-labeling and hyper-parameter tuning. They evaluate their method on IPC domains and a robotic domain, and also perform an ablation study.

**Strengths:**

The paper is looking at an important problem, and something that can't currently be solved by simply calling an LLM. The automatic generation of valid symbolic models can allow the use of provably sound planners in many mission-critical settings. The evaluation also shows that the proposed system does provide some advantages over some existing alternatives.

**Weaknesses:**

The method described here is just relying on repeated invocation of LLMs to perform refinement and model generation (and in the case of pseudo-labeling, a VLM). The hope is that, given the right feedback, the LLM should be able to find the right models and symbols. However, apart from the limited empirical evidence they can show, I cannot imagine the approach being able to provide any kind of  theoretical guarantees.

More importantly, the paper completely overlooks many of the most important aspects of hierarchical planning, many of which are quite relevant to the paper at hand. It is very unfortunate that the authors have chosen to overlook all of them and have tried to rely on their own simplistic notions of hierarchical planning. To start with, I would recommend that the authors start by looking at hierarchical task networks. One could see a textbook-level introduction to these models in most introductory planning books, but the book “Automated Planning and Acting” [1] is an excellent starting point. A particularly important notion that shows up within this context is the idea that a higher-level task could be refined into multiple possible sequences of atomic actions. While the planning literature also points to the notion of macros [2], where each macro refines into a single unique action sequence, they cannot capture the kind of hierarchies the paper alludes to. For example, if one wants to capture an abstraction action of picking up an object. The refinement of this action could potentially involve multiple kinds of grasp positions, and as such, corresponds to different kinds of refinements. However, once you have such refinements, a subset relationship isn’t enough. You would need more complex mechanisms to capture the semantics of the intermediate actions. Here are some papers that provide some sample methods to associate semantics to such abstractions [3,4].

I believe the paper looks at an interesting problem, but the work itself is quite preliminary, and the authors seem to overlook a lot of foundational work in hierarchical planning. I would recommend that the authors read up on this larger body of literature and try to place their work in this broader context.

Smaller comments:
The paper keeps talking about interpretability. However, this is never justified or expounded upon. I would probably leave out claims like planning leads to “interpretable plans”, particularly from the motivation part of the work.

[1] Ghallab, Malik, Dana Nau, and Paolo Traverso. Automated planning and acting. Cambridge University Press, 2016.

[2] Botea, Adi, Martin Müller, and Jonathan Schaeffer. "Learning partial-order macros from solutions." Proceedings of the Fifteenth International Conference on Automated Planning and Scheduling. 2005.

[3] Marthi, Bhaskara, Stuart Russell, and Jason Andrew Wolfe. "Angelic Semantics for High-Level Actions." ICAPS. 2007.

[4] Srivastava, Siddharth, Stuart Russell, and Alessandro Pinto. "Metaphysics of planning domain descriptions." Proceedings of the AAAI Conference on Artificial Intelligence. Vol. 30. No. 1. 2016.

**Questions:**

I would request the authors to respond to my concerns regarding the lack of theoretical guarantees and the inability to support multiple refinements of higher-level actions.

---

> ### Author Response · Authors · 2025-11-21
>
> We thank you for the review and the constructive feedback and want to address the stated questions and weaknesses in the following:
>
> **Regarding Hierarchical Planning**
> We acknowledge that HTN planning is relevant and have added a dedicated section in the Related Work. However, there is a key distinction: HTN assumes a given hierarchical structure and focuses on planning within it, whereas LODGE learns the hierarchical domain model itself.
> In LODGE, the complexity of large domains is handled by splitting learning across multiple levels. Each level uses a smaller, more focused model, and high-level actions are treated like regular actions that can be planned with standard planners. This also allows LODGE to decide automatically when and how to decompose tasks, rather than relying on predefined hierarchies.
> In short, while HTN planning and LODGE both use hierarchical ideas, LODGE addresses the fundamentally different problem of learning hierarchical models, not planning with them.
>
>
> **Regarding different grasp positions**
> LODGE is concerned about learning hierarchical domain models for abstract planning. Planning correct grasp positions in this context can be ensured in two ways: (i) treat the grasp pose as a continuous parameter and (ii) inventing a predicate for special grasps.
> - (i) Continuous skill parameters, such as grasp positions, are generally not treated on the task planner's level (see [A]). Instead, continuous parameters are determined when translating the abstract plan into a concrete one, e.g. via learned samplers or grounding methods for the continuous parameters. These samplers or grounding methods are naturally compatible with LODGE. Samplers can still be used if the grasp pose influences the high-level plan success, e.g. via bilevel planning (see [A]).
> - (ii) When the grasp pose influences the success of the higher-level plan, it could be helpful to generate a specific `grasp`-predicate for it. This `grasp`-predicate ensures the objects is grasped such that the high-level plan succeeds, which would prevent bilevel planning.
>
> **Regarding 'Repeated Invocations of LLMs'**
> We fully agree that theoretical guarantees would be highly desirable. The fundamental challenge we address is leveraging the implicit common knowledge encoded in LLMs, which is essential for autonomously inferring task-relevant domain models, while simultaneously mitigating their inherent weaknesses (lack of explicit knowledge representation, hallucinations). Due to these limitations, and the assumption that no ground truth is available (if it were, we wouldn't need the LLM), obtaining strong correctness guarantees seems impossible. We argue that the consequence of this should be neither to blindly trust the LLM, nor to dismiss this powerful source of information completely. Instead, we propose to systematically validate and correct the outputs by checking for inconsistencies. Namely, we designed an empirically validated automatic process to detect inconsistencies between simulation and domain model, and also within different abstractions within the domain model, and to resolve these inconsistencies. Dismissing this as mere re-prompting is not fair, in particular given that there currently doesn't seem to be any clear path towards obtaining common knowledge in a verifyable way.
>
> **Regarding 'Interpretability'**
> We understand that 'interpretability' should be used carefully. However, by using PDDL as language to learn explicit domain models, the resulting PDDL domain, and the later plan, are represented in that language, and can be interpreted directly, or even translated into natural language [B,C]. We therefore believe that using interpretability  as motivation is valid, there also exists a body of work that tries to provide explainability for AI planners based on their interpretability (e.g. [D]). In contrast, end-to-end approaches such as VLAs translate instructions directly into robot joint actions and do not produce explicit, interpretable symbolic models.
>
>
> - [A] T. Silver et al., “Predicate invention for bilevel planning,”. https://ojs.aaai.org/index.php/AAAI/article/view/26429
> - [B] L. Guan, K. Valmeekam, S. Sreedharan, and S. Kambhampati, “Leveraging Pre-trained Large Language Models to Construct and Utilize World Models for Model-based Task Planning,”
> - [C] S. Sreedharan and S. Kambhampati, “Leveraging pddl to make inscrutable agents interpretable: A case for post hoc symbolic explanations for sequential-decision making problems,”
> - [D] M. Fox, D. Long, and D. Magazzeni, “Explainable Planning,”

---

### Author Response · Authors · 2025-11-27

Dear Reviewers,
we wanted to follow up regarding our rebuttal and kindly ask whether our responses addressed all of your concerns. Please let us know if there is anything further we can clarify.
Thank you for your time.

---

### Author Response · Authors · 2025-12-03
**Rebuttal Summary for the AE**

We thank all reviewers for their constructive feedback during the rebuttal. Below, we summarize the key changes made in response to their concerns; all additions are highlighted in blue in the manuscript.

**Additional Experiments (Reviewers 3u2N, MNNh, p9vb):** We significantly expanded our experimental evaluation, particularly on FurnitureBench:

- **Simulation experiments:** We demonstrated generalization by (1) adapting the learned Lamp domain to assemble the Round Table, and (2) learning a Square Table domain model and successfully applying it to assemble the Stool and Desk.
- **Real-world experiments:** We added two new real-robot experiments on a Franka system. First, we demonstrated that domain models learned in simulation transfer reliably to real-world planning and execution (video included in supplementary materials). Second, addressing a key concern from reviewers MNNh and p9vb, we showed that LODGE can learn hierarchical domain models directly from real-world interactions despite increased noise and variance. The real-world learned model achieved 38% task success, substantially outperforming all baseline methods even when they use skill annotations.

These experiments demonstrate that LODGE's approach of combining hierarchical domain models with targeted environment feedback and LLM refinement enables learning accurate models from few environment interactions in both simulated and real settings.

**Positioning and Theoretical Context (Reviewer PUwn):** Following reviewer PUwn's feedback, we added a dedicated Related Work section on hierarchical planning (highlighted in blue). In comparison to classical HTN planning, which assumes predefined hierarchies, our contribution is about learning hierarchical domain models. We clarify that LODGE automatically determines *when* and *how* to decompose tasks, distributing learning complexity across abstraction levels while enabling classical planners to operate at each level. We also added an ablation (Table 7) demonstrating that hierarchical domain models significantly improve both domain accuracy (0.78 vs 0.30 EW) and task success (0.81 vs 0.13) compared to flat domain learning.

**LLM Usage Analysis (Reviewer 3u2N):** We added a ablation (Appendix A.1, Table 6) comparing token consumption and number of LLM calls across all methods. LODGE achieves domain model learning with a smaller token consumption, compared to related work.

**Clarifications on Novelty and Design Choices:** We addressed reviewer MNNh's concerns about our three main contributions by clarifying that (1) LODGE requires minimal prior knowledge (only goal predicates and object types, consistent with prior work), while learning all operators and additional predicates autonomously; (2) our hierarchical formulation distributes the complexity of learning the domain model without increasing planning cost; and (3) our learning of predicate classifiers combines VLM pseudo-labeling with code-based classifiers, which avoids a VLM in-the-loop while achieving higher accuracy than related methods.

---

### Meta-Review · Area_Chair_s31Q · 2026-01-07

**Summary:**

This paper addresses an important problem: autonomously learning hierarchical symbolic planning domain models by leveraging LLMs and environment feedback. The authors have been responsive to reviewer feedback, substantially strengthening the paper through additional evaluations, ablations, and real-robot experiments.

That said, I believe the paper would benefit from greater clarity and better positioning.

In particular, I would encourage the authors to more explicitly articulate their hierarchical operator formulation, including:
- How abstract operators relate to lower-level ones in terms of downward refinability
- What assumptions are made (implicitly or explicitly) for downward refinement to hol
- How this formulation connects to classical hierarchical planning and HTN literature, where downward refinability has been studied extensively, including settings with formal guarantees

This clarification is especially important in real-world robotic domains, where downward refinability is often not guaranteed due to tight coupling between continuous parameters such as grasp poses, placement poses, and contact modes.

Relatedly, since the paper emphasizes hierarchical planning and hierarchical domain learning, I would encourage the authors to consider domains with more inherently hierarchical structure. In the current evaluations, the tasks are relatively shallow hierarchically, making it difficult to assess when and why hierarchical operator learning provides clear advantages over flat domain learning (and whether the gain of LODGE is mainly due to better usage of VLMs for pseudo-labels / iterative self-criticism, etc.).

**Reviewer Concerns:**

Outstanding:
- Relation to Hierarchical Planning / HTN. The authors have added additional references. However, I think the authors are still missing important acknowledgements to
  1) In HTN, one can also assume downward refinability to accelerate the planning procedure. And there are also algorithms with theoretical guarantees on "greedily apply downward refinability and backtrack when needed". See: https://people.csail.mit.edu/lpk/papers/hpnICRA11Final.pdf
  2) There are also papers on "learning HTNs," although of course most of them do not use LLMs.
- Empirical Validation of Hierarchy Benefits:
  Compared to many cited work on leveraging VLMs/LLMs for predicate invention, it seems that the main contribution of this paper is the added hierarchical learning part. However, on the domains evaluated, it is unclear how hierarchical operator learning is really helpful here since the tasks are not very "hierarchical."
- End-to-end planning success vs InterPreT, Pix2Pred, etc.

Addressed:
- FurnitureBench Coverage
- LLM / Token Usage & Efficiency
- Low-Level Control Failures vs Model Errors
- Real Robot Experiments
- Clarifications on VLM Pseudo-Labels

**Reviewer Scores:**

Reviewer PUwn would possibly not update their score because the main concern about problem formulation in hierarchical planning is not fully addressed (see my "reviewer concerns" part)
Reviewer 3u2N has updated their score.
Reviewer MNNh would likely slightly raise their score because their questions were partially addressed but not all (e.g., end-to-end evaluation of InterPreT, Pix2Pred)
Reviewer p9vb would likelly not raise their score because there is no clear ablation and justification of their handling of VLM Pseudo-Labels.

---

### Decision · Program_Chairs · 2026-01-26

Reject